# LEARNING SYMMETRIES THROUGH LOSS LANDSCAPE

## ABSTRACT

Incorporating equivariance as an inductive bias into deep learning architectures to take advantage of the data symmetry has been successful in multiple applications, such as chemistry and dynamical systems. In particular, roto-translations are crucial for effectively modeling geometric graphs and molecules, where understanding the 3D structures enhances generalization. However, equivariant models often pose challenges due to their high computational complexity. In this paper, we introduce REMUL, a training procedure for approximating equivariance with multitask learning. We show that unconstrained models (which do not build equivariance into the architecture) can learn approximate symmetries by minimizing an additional simple equivariance loss. By formulating equivariance as a new learning objective, we can control the level of approximate equivariance in the model. Our method achieves competitive performance compared to equivariant baselines while being $10\times$ faster at inference and $2.5\times$ at training.

## 1 INTRODUCTION

Equivariant machine learning models have achieved notable success across various domains, such as computer vision (Weiler et al., 2018; Yu et al., 2022), dynamical systems (Han et al., 2022; Xu et al., 2024), chemistry (Satorras et al., 2021; Brandstetter et al., 2022), and structural biology (Jumper et al., 2021). For example, incorporating equivariance *w.r.t.* translations and rotations ensures the correct handling of complex structures like graphs and molecules (Schütt et al., 2021; Bronstein et al., 2021; Thölke & Fabritiis, 2022; Liao et al., 2024). Equivariant machine learning models benefit from this inductive bias by *explicitly* leveraging symmetries of the data during the architecture design. Typically, such architectures have highly constrained layers with restrictions on the form and action of weight matrices and nonlinear activations (Batzner et al., 2022; Batatia et al., 2022). This may come at the expense of higher computational cost, making it sometimes challenging to scale equivariant architectures, particularly those relying on spherical harmonics and irreducible representations (Thomas et al., 2018; Fuchs et al., 2020; Liao & Smidt, 2023; Luo et al., 2024). On the other hand, equivariance constraints might limit the expressive power of the network, restricting its ability to act as a universal architecture (Dym & Maron, 2021; Joshi et al., 2023).

Equivariant layers are not the only way to incorporate symmetries into deep neural networks. Several approaches have been proposed to either offload the equivariance restrictions to faster networks (Kaba et al., 2022; Mondal et al., 2023; Baker et al., 2024; Ma et al., 2024; Panigrahi & Mondal, 2024) or simplify the constraints by introducing averaging operations (Puny et al., 2022; Duval et al., 2023; Lin et al., 2024; Huang et al., 2024). Nonetheless, while these approaches leverage unconstrained architectures, they often require additional networks or averaging techniques to achieve equivariance and may not rely solely on adjustments to the training protocol. To this aim, a widely adopted strategy to replace 'hard' equivariance (i.e., built into the architecture itself) with a 'soft' one, is *data augmentation* (Quiroga et al., 2019; Bai et al., 2021; Gerken et al., 2022; Iglesias et al., 2023; Xu et al., 2023; Yang et al., 2024), whereby the training protocol of an arbitrary (unconstrained) network is augmented by assigning the same label to group orbits (e.g., rotated and translated versions of the input). In fact, recent works have shown that unconstrained architectures may offer a valid alternative provided that enough data are available (Wang et al., 2024; Abramson et al., 2024).

Besides the challenges in computational cost and design, there are also tasks that do not exhibit full equivariance, such as dynamical phase transitions (Baek et al., 2017; Weidinger et al., 2017), polar fluids (Gibb et al., 2024), molecular nanocrystals (Yannouleas & Landman, 2000), and cellular

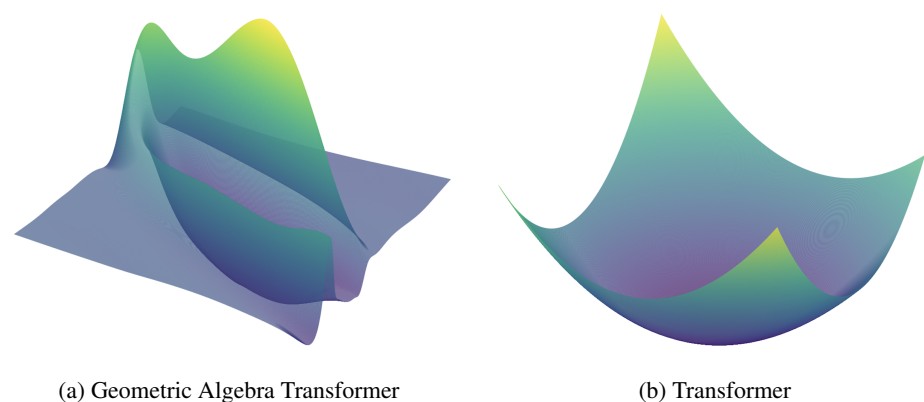

(a) Geometric Algebra Transformer            (b) Transformer

Figure 1: Loss surface around local minima of trained models on N-body dynamical system.

symmetry breaking (Goehring et al., 2011; Mietke et al., 2019). For such tasks, fully-equivariant networks might be excessively constrained, which further motivates the design of a more flexible approach.

In this work, we present **REMUL**: **R**elaxed **E**quivariance via **Mu**ltitask **L**earning. REMUL is a training procedure that aims to learn approximate equivariance during training for unconstrained networks using a multitask approach with adaptive weights. We conduct a comprehensive evaluation of unconstrained models trained with REMUL, comparing their performance and computational efficiency against equivariant models. We consider Transformers and Graph Neural Networks (GNNs), as well as their E(3) equivariant versions, as our main baselines, focusing on roto-translation group.

Our contributions are as follows:

- We formulate equivariance as a weighted multitask learning objective for unconstrained models, aiming to simultaneously learn the objective function and approximate the required equivariance associated with the data and the task.

- We demonstrate that by adjusting the weighting of the equivariance loss, we can modulate the extent to which a model exhibits equivariance, depending on the requirements of the task. Specifically, tasks that demand full equivariance require a higher weight on the equivariance component, whereas tasks that require less strict equivariance can be managed with lower weights.

- Empirically, we show that Transformers and Graph Neural Networks trained with our multitask learning approach compete or outperform their equivariant counterparts.

- By leveraging the efficiency of Transformers, we achieve up to $10\times$ speed-up at inference and $2.5\times$ speed-up in training compared to equivariant baselines. This finding could provide motivations for the use of unconstrained models, which do not require equivariance in their design, potentially offering a more practical approach.

- We point out that the standard Transformer exhibits a more convex loss surface near the local minima compared to the Geometric Algebra Transformer (Brehmer et al., 2023), which can indicate further evidence of the optimization difficulties of equivariant networks.

## 2 BACKGROUND

### 2.1 SYMMETRY GROUPS AND EQUIVARIANT MODELS

Symmetry groups, a fundamental concept in abstract algebra and geometry, are a mathematical description of the properties of an object remaining unchanged (invariant) under a set of transformations. Formally, a symmetry group $G$ of a set $X$ is a group of bijective functions from $X$ to itself, where the group operation is function composition.

Equivariant machine learning models are designed to preserve the symmetries associated with the data and the task. In geometric deep learning (GDL), the data is typically assumed to live on some

geometric domain (e.g., a graph or a grid) that has an appropriate symmetry group (e.g., permutation or translation) associated with it. Equivariant models implement functions $f : X \to Y$ from input domain $X$ to output domain $Y$ that ensure the actions of a symmetry group $G$ on data from $X$ correspond systematically to its actions on $Y$, through the respective group representations $\phi$ and $\rho$. Formally, we say that:

**Definition 2.1.** *A function $f$ is* equivariant *w.r.t. the group $G$ if for any transformation $g \in G$ and any input $x \in X$,*

$$f(\phi(g)(x)) = \rho(g)(f(x)) \tag{1}$$

The group representations $\phi$ and $\rho$ allow us to apply abstract objects (elements of the group $G$) on concrete input and output data, in the form of appropriately defined linear transformations. For example, if $G = S_n$ (a permutation group of $n$ elements, arising in learning on graphs with $n$ nodes), its action on $n$-dimensional vectors (e.g., graph node features or labels) can be represented as an $n \times n$ permutation matrix.

A special case of equivariance is obtained for a trivial output representation $\rho = \mathrm{id}$:

**Definition 2.2.** *A function $f$ is* invariant *w.r.t. the group $G$ if for all $g \in G$ and $x \in X$,*

$$f(\phi(g)(x)) = f(x) \tag{2}$$

## 2.2 Equivariance as a Constrained Optimization Problem

Consider a class of parametric functions $f_\theta$, typically implemented as neural networks, whose parameters $\theta$ are estimated via a general training objective based on data pairs $(x, y) \sim q$:

$$\underset{\theta}{\text{minimize}} \quad \mathbb{E}_{(x,y)\sim q} \left[ \mathcal{L}(f_\theta(x), y) \right] \tag{3}$$

Here, $\mathcal{L}$ represents the loss function that quantifies the discrepancy between the model's predictions $f_\theta(x)$ and the true labels $y$. The class of models is considered equivariant with respect to a group $G$ if it satisfies the constraint in Equation 1 for any input $x \in X$ and for any action $g \in G$.

Equivariance is typically achieved *by design*, by imposing constraints on the form of $f_\theta$. Since $f_\theta$ is usually composed of multiple layers, ensuring equivariance implies restrictions on the operations performed in each layer, a canonical example being message-passing graph neural networks whose local aggregations need to be permutation-equivariant to respect the overall invariance to the action of the symmetric group $S_n$. As such, finding an equivariant solution to the minimization problem in Equation 3 corresponds to solving the following constrained optimization:

$$\begin{aligned} \underset{\theta}{\text{minimize}} \quad & \mathbb{E}_{(x,y)\sim q} \left[ \mathcal{L}(f_\theta(x), y) \right] \\ \text{subject to} \quad & f_\theta(\phi(g)(x)) = \rho(g) f_\theta(x), \ \forall g \in G, \ \forall x \in X \end{aligned} \tag{4}$$

In general, such optimization is challenging, leading to complex design choices to enforce equivariance that could ultimately restrict the class of minimizers and make the training harder. Additionally, for relevant tasks, the optimal solution only needs to be approximate equivariant (Wang et al., 2022; Petrache & Trivedi, 2023; Kufel et al., 2024; Ashman et al., 2024) meaning that the extent to which a model needs to exhibit equivariance can vary significantly based on the specific characteristics of the data and the requirements of the downstream application. In light of these reasons, we necessitate a flexible approach to incorporating equivariance into the learning process. To address this, we propose REMUL, a training procedure that replaces the hard optimization problem with a soft constraint, by using a multitask learning approach with adaptive weights.

## 3 Learning Symmetries through Loss Landscape

### 3.1 Equivariance as a New Learning Objective

Our main idea is to formulate equivariance as a multitask learning problem for an unconstrained model. We achieve that by *relaxing* the optimization problem in Equation 4. Namely, once we introduce a functional $\mathcal{F}_{\mathcal{X},G}$ that measures the equivariance of a candidate function $f_\theta$, we replace the constrained variational problem in Equation 4 with

$$\underset{\theta}{\text{minimize}} \ \mathbb{E}_{(x,y)\sim q} \left[ \alpha \mathcal{L}(f_\theta(x), y) + \beta \mathcal{F}_{\mathcal{X},G}(f_\theta(x), y) \right], \tag{5}$$

where $\alpha, \beta > 0$. This decomposition allows for tailored learning dynamics where the supervised loss specifically addresses the information from the dataset without constraining the solution $f_\theta$, while the equivariance penalty $\mathcal{F}$ smoothly enforces symmetry preservation.

We note that in conventional supervised settings, one has access to a dataset $\mathcal{X} = \{x_1, x_2, \ldots, x_n\}$ with corresponding labels $\mathcal{Y} = \{y_1, y_2, \ldots, y_n\}$. We can then introduce

$$\mathcal{L}_{\text{obj}}(f_\theta, \mathcal{X}, \mathcal{Y}) = \sum_{i=1}^{n} \mathcal{L}(f_\theta(x_i), y_i), \tag{6}$$

and formulate the optimization as:

$$\mathcal{L}_{\text{total}}(f_\theta, \mathcal{X}, \mathcal{Y}, G) = \alpha \mathcal{L}_{\text{obj}}(f_\theta, \mathcal{X}, \mathcal{Y}) + \beta \mathcal{L}_{\text{equi}}(f_\theta, \mathcal{X}, \mathcal{Y}, G), \tag{7}$$

where $\mathcal{L}_{\text{equi}}(f_\theta, \mathcal{X}, \mathcal{Y}, G)$ represents our *augmented equivariance loss*, specifically designed to enforce the model's adherence to the symmetry action of the group $G$, given a dataset $\mathcal{X}$ and labels $\mathcal{Y}$. For a finite number of training samples $n$, we propose an equivariant loss $\mathcal{L}_{\text{equi}}$ of the form:

$$\mathcal{L}_{\text{equi}}(f_\theta, \mathcal{X}, \mathcal{Y}, G) = \sum_{i=1}^{n} \ell(f_\theta(\phi(g_i)(x_i)), \rho(g_i)(y_i)) \tag{8}$$

Here $\ell$ is a metric function, typically an $L_1$ or $L_2$ norm, that quantifies the discrepancy between $f(\phi(g_i)(x_i))$ and $\rho(g_i)(y_i)$, with $g_i \in G$ randomly-selected group elements for each sample. In fact, in our implementation, we change the group elements being sampled in each training step.

The parameters $\alpha$ and $\beta$ defined in Equation 7 are weighting factors that balance the traditional objective loss with the equivariance loss, enabling practitioners to tailor the training process according to specific requirements of symmetry and generalization. More specifically, a large value of $\beta$ indicates a more equivariant function while the smaller value of $\beta$ indicates a less equivariant function. These parameters allow us to control the trade-off between model generalization and equivariance, based on the specific requirements of the task.

## 3.2 ADAPTING PENALTY PARAMETERS DURING TRAINING

For simultaneously learning the objective and equivariance losses, we consider two distinct approaches to regulate the penalty parameters $\alpha$ and $\beta$: *constant* penalty and *gradual* penalty. The constant penalty assigns a fixed weight to each task's loss throughout the training process. In contrast, the gradual penalty dynamically adjusts the weights of each task's loss during training. For gradual penalty, we use the GradNorm algorithm introduced by Chen et al. (2018), which is particularly suited for tasks that involve simultaneous optimization of multiple loss components, as it dynamically adjusts the weight of each loss during training. It updates the weights of the loss components based on the magnitudes of their gradients, *w.r.t* the last layer in the network, which is essential for the contribution of each loss. It also has a learning rate parameter $\eta$, that fine-tunes the speed at which the weights are updated, providing precise control over their convergence rates (see Algorithm 1 for details).

---

**Algorithm 1** GradNorm Algorithm (one step)

---

1: **Input:** $\alpha$, $\beta$, $\eta$, $\gamma$, $\mathcal{L}_{\text{obj}}$, $\mathcal{L}_{\text{equi}}$, and $\mathcal{W}$ (the weights of the last layer in the network)
2: $\mathcal{G}_{\text{obj}} = \|\nabla_{\mathcal{W}} \alpha \mathcal{L}_{\text{obj}}\|_2$, $\tilde{\mathcal{L}_{\text{obj}}} = \mathcal{L}_{\text{obj}}/\mathcal{L}_{\text{obj}}(0)$
3: $\mathcal{G}_{\text{equi}} = \|\nabla_{\mathcal{W}} \beta \mathcal{L}_{\text{equi}}\|_2$, $\tilde{\mathcal{L}_{\text{equi}}} = \mathcal{L}_{\text{equi}}/\mathcal{L}_{\text{equi}}(0)$
4: $\bar{\mathcal{G}} = \frac{\mathcal{G}_{\text{obj}} + \mathcal{G}_{\text{equi}}}{2}$, $r = \frac{\tilde{\mathcal{L}_{\text{obj}}} + \tilde{\mathcal{L}_{\text{equi}}}}{2}$
5: $r_\alpha = \frac{\tilde{\mathcal{L}_{\text{obj}}}}{r}$, $r_\beta = \frac{\tilde{\mathcal{L}_{\text{equi}}}}{r}$
6: $\mathcal{L}_{\text{g}} = |\mathcal{G}_{\text{obj}} - \bar{\mathcal{G}} \times [r_\alpha]^\gamma| + |\mathcal{G}_{\text{equi}} - \bar{\mathcal{G}} \times [r_\beta]^\gamma|$
7: $\alpha = \alpha - \eta \nabla_\alpha \mathcal{L}_{\text{g}}$
8: $\beta = \beta - \eta \nabla_\beta \mathcal{L}_{\text{g}}$
9: **Return:** $\alpha, \beta$

---

### 3.3 EQUIVARIANCE WITH DATA AUGMENTATION

Data augmentation is a widely recognized technique that enhances the performance of machine learning models by including different transformations in the training process. It involved creating a transformed input and measuring the original loss between the model prediction and the transformed target. In contrast, our method utilizes an additional *controlled* equivariance loss to incorporate symmetrical considerations simultaneously with the objective loss during training. In fact, traditional data augmentation techniques can be interpreted as special cases of Equation 7 where $\alpha = 0$ and $\beta = 1$.

## 4 QUANTIFYING LEARNED EQUIVARIANCE

Using group transformations to measure and assess the symmetries of ML models has been studied in several domains (Lyle et al., 2020; Kvinge et al., 2022; Moskalev et al., 2023; Gruver et al., 2023; Speicher et al., 2024). Inspired by the idea of frame-averaging (Puny et al., 2022; Duval et al., 2023; Lin et al., 2024), in this section, we introduce a metric to quantify the degree of equivariance exhibited by a function $f$.

Starting from Equation 1, the group integration of both sides *w.r.t.* the normalized Haar measure $\mu$ yields:

$$\int_G f(\phi(g)(x)) \, d\mu(g) = \int_G \rho(g)(f(x)) \, d\mu(g) \tag{9}$$

When $G$ is a large or continuous group, as is the case in our work, the integrals over $G$ may not be computable in closed form. Therefore, we approximate the integrals using a Monte Carlo approach with samples $\{g_i\}_{i=1}^M$ from $G$:

$$\int_G f(\phi(g)(x)) \, d\mu(g) \approx \frac{1}{M} \sum_{i=1}^M f(\phi(g_i)(x)) \tag{10}$$

$$\int_G \rho(g)(f(x)) \, d\mu(g) \approx \frac{1}{M} \sum_{i=1}^M \rho(g_i)(f(x)) \tag{11}$$

Where $M$ is a large number of samples from $G$. Given the group averages, we define the equivariance error $E(f, G)$ as the average norm of the difference between these two averages over the data distribution $D$:

$$E(f, G) = \frac{1}{|D|} \sum_{x \in D} \left\| \frac{1}{M} \sum_{i=1}^M \rho(g_i)(f(x)) - \frac{1}{M} \sum_{i=1}^M f(\phi(g_i)(x)) \right\|_2 \tag{12}$$

Here $\| \cdot \|_2$ denotes an $L_2$ norm (for non-scalar function). This error indicates the average deviation of a function $f$ from perfect equivariance across the data distribution $D$ (lower value means more equivariant function).

We also propose another measure that takes the average over the group of differences between $f(\phi(g)(x))$ and $\rho(g)(f(x))$,

$$E'(f, G) = \frac{1}{|D|} \sum_{x \in D} \frac{1}{M} \sum_{i=1}^M \| f(\phi(g_i)(x)) - \rho(g_i)(f(x)) \|_2 \tag{13}$$

Equation 12 & Equation 13 indicate a practical metric for evaluating how closely the function $f$ approximates perfect equivariance throughout a data distribution $D$ (which should be zero for a perfect equivariance function). In practice, we use $M = 100$ samples from the group and noticed this was sufficient to obtain stable results. We also observed that both measures have very similar behavior in our experiments, where $E$ and $E'$ are near zero for equivariant models. We also demonstrate that increasing the value of $\beta$ in Equation 7 results in a less equivariant error for $E$ and $E'$.

## 5 RELATED WORK

**Equivariant ML Models.** In the vision domain, group convolutions have proven to be a powerful tool for handling rotation equivariance for images and enhanced model generalization (Cohen & Welling, 2016; Cohen et al., 2019; Weiler & Cesa, 2019; Qiao et al., 2023). Similarly, the development of equivariant architectures with respect to roto-translations for geometric data has been an active area of research (Chen et al., 2021a; Satorras et al., 2021; Han et al., 2022; Xu et al., 2024). Techniques that use spherical harmonics and irreducible representations have shown a large success in modeling graph-structured data, such as SE(3)-Transformers (Fuchs et al., 2020), Tensor Field Networks (Thomas et al., 2018), and DimeNet (Gasteiger et al., 2020). More recently, Brehmer et al. (2023) introduced an E(3) equivariant Transformer that employs geometric algebra for processing 3D point clouds.

**Data Augmentation and Unconstrained Models.** Alternatively, integrating transformations through data augmentation is a widely used strategy across multiple vision tasks, enhancing performance in image classification (Perez & Wang, 2017; Inoue, 2018; Rahat et al., 2024), object detection (Zoph et al., 2020; Wang et al., 2019; Kisantal et al., 2019), and segmentation (Negassi et al., 2022; Chen et al., 2021b; Yu et al., 2023). For geometric data, Hu et al. (2021) has adapted a Graph Neural Network architecture with data augmentation to process 3D molecular structures. In parallel, Dosovitskiy et al. (2021) introduced that Vision Transformers (ViTs) with a large amount of training data can achieve comparable performance with Convolutional Neural Networks (CNNs), obviating the need for explicit translation equivariance within the architecture. Recently, this has shown to be effective for handling geometric data (Wang et al., 2024; Abramson et al., 2024).

**Learning Symmetries and Approximate Equivariance.** Several studies have shown that the layers of CNN architectures can be approximated for a soft constraint (Wang et al., 2022; van der Ouderaa et al., 2022; Romero & Lohit, 2022; Veefkind & Cesa, 2024; Wu et al., 2024; McNeela, 2023). Conversely, van der Ouderaa et al. (2023) extends the Bayesian model selection approach to learning symmetries in image datasets. Yeh et al. (2022) introduced a parameter-sharing scheme to achieve permutations and shifts equivariances in Gaussian distributions. Recent works have relaxed the hard constrained models to a soft constraint by adding unconstrained layers in the architecture design (Finzi et al., 2021a; Pertigkiozoglou et al., 2024), canonicalization network (Lawrence et al., 2024). , or explicit relaxation Kaba & Ravanbakhsh (2023). Additionally, Lin et al. (2019) modified the loss of CNN for segmentation task. Shakerinava et al. (2022) introduced a method to learn equivariant representation using the group invariants, while Bhardwaj et al. (2023) defined a regularizer that injects the equivariance in the latent space of the network by explicitly modeling transformations with additional learnable maps. In contrast, several works have started from pre-trained models (Basu et al., 2023; Kim et al., 2023b). Furthermore, the EGNN framework (Satorras et al., 2021) has been modified using an invariant function (Zheng et al., 2024) or adversarial training procedure (Yang et al., 2023). However, in our work, we start completely from unconstrained models without assuming any equivariance over the space of functions in the architecture design. Moreover, we didn't assume a specific class of models or introduce additional parameters, which increases the applicability of our method to various domains and makes it computationally efficient.

## 6 EXPERIMENTS AND DISCUSSION

In this section, we aim to compare constrained equivariant models with unconstrained models trained with REMUL, our multitask approach. We are targeting three main questions: Can unconstrained models learn the approximate equivariance, how does that affect the performance & generalization, and what are their computational costs. We evaluate our method on different tasks for geometric data: N-body dynamical system (Section 6.1), motion capture (Section 6.2), and molecular dynamics (Section 6.3). For unconstrained models, we apply REMUL to Transformers and Graph Neural Networks. We then compare against their equivariant baselines: SE(3)-Transformer (Fuchs et al., 2020), Geometric Algebra Transformer (Brehmer et al., 2023), and Equivariant Graph Neural Networks (Satorras et al., 2021) as well as unconstrained models with data augmentation. We consider learning the rotation group $SO(3)$ for REMUL and data augmentation and we subtract the center of mass for translation. We use the equivariance metric defined in Equation 12 to analyze our results. We also conduct a comparative analysis for the computational requirements of unconstrained models and equivariant models in Section 6.4. Lastly, we discuss the loss surfaces in Section 6.5. Implementation details and additional experiments can be found in Appendix B & Appendix C.

## 6.1 N-Body Dynamical System

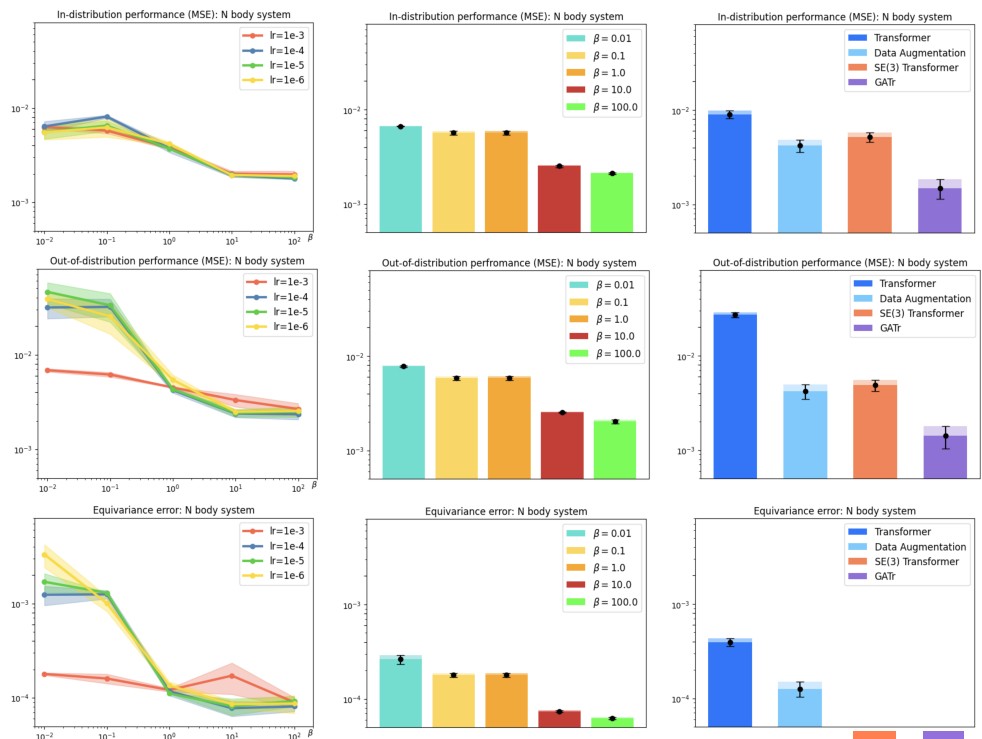

Figure 2: N-body dynamical system. Each row represents a different evaluation scenario: The top row shows in-distribution performance, the middle row displays out-of-distribution performance, and the bottom row illustrates equivariance error. The columns correspond to different architectures/ model conditions (from left to right): The first column shows the Transformer trained with REMUL (gradual penalty), the second column with a constant penalty, and the third column presents the baselines (equivariant models, standard Transformer, and data augmentation). The equivariance metric included in this Figure is defined in Equation 12, we report the same plots for the metric defined in Equation 13 in Appendix C.1 (Figure 6), which has a similar behavior. Transformer architecture with high $\beta$ reduces the equivariance error and improves the performance. SE(3)-Transformer and GATr have a small equivariance error below the range of the plots ($2.8e^{-10}$ and $1.13e^{-15}$ respectively).

To conduct ablation studies of our method, we utilized the dynamical system problem described by Brehmer et al. (2023). The task involves predicting the positions of particles after $100$ Euler time steps of Newton's motion equation, given initial positions, masses, and velocities. This problem is inherently equivariant under rotation and translation groups, implying that any rotation/translation of the initial states should rotate/translate the final states of the particles by the same amount. We conduct comparisons between Transformer trained with REMUL against two equivariant architectures: SE(3)-Transformer and Geometric Algebra Transformer (GATr). We use the same Transformer version and hyperparameters specified by Brehmer et al. (2023). Additional implementation details, including in-distribution and out-of-distribution settings, are provided in Appendix B.1. Our results are presented in Figure 2.

From Figure 2, we noticed that increasing the penalty parameter $\beta$ of the equivariance loss significantly reduces the equivariance error in both constant and gradual settings (which results in a more equivariant model). Equivariant architectures demonstrate an equivariance error near zero, which is expected by their design. The performance behaves similarly; a higher penalty enhances model generalization for both in-distribution and out-of-distribution. Transformer with high $\beta$ outperforms both data augmentation and SE(3)-Transformer across in-distribution and out-of-distribution and competes with GATr. We also observe that despite SE(3)-Transformer having a substantially lower equivariance error, its performance is slightly worse than Transformer trained with data augmenta-

tion. This highlights that equivariance, although improving generalization in this task, is only one aspect of understanding model performance. Lastly, the standard Transformer (without REMUL and data augmentation) exhibits the highest equivariance error and the lowest overall performance.

## 6.2 MOTION CAPTURE

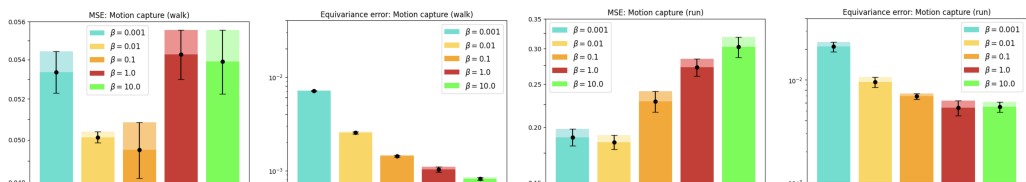

Figure 3: Motion Capture dataset: Transformer trained with REMUL. Two figures on the left: Performance (MSE) and equivariance error for walking task (Subject #35), respectively. Two figures on the right: Performance (MSE) and equivariance error for running task (Subject #9), respectively. We use the equivariance metric described in Equation 12 and include the same plots for the second metric (Equation 13) in Appendix C.3 (Figure 8). We show a trade-off between model performance and equivariance error, where high penalty $\beta$ gives less equivariance error (more equivariant model) but the best performance comes at an intermediate level of equivariance for both tasks.

Table 1: Performance on Motion Capture dataset: MSE ($\times 10^{-2}$). REMUL procedure and data augmentation were applied to standard Transformer & MLP. First, Second (highlighted). REMUL comes the best in both tasks.

|  | SE(3)-Transformer | GATr | Transformer | Data Augmentation | Ours |
|---|---|---|---|---|---|
| Walking (Subject #35) | $10.85_{\pm 1.3}$ | $10.06_{\pm 1.3}$ | $5.21_{\pm 0.08}$ | $5.3_{\pm 0.18}$ | $4.95_{\pm 0.1}$ |
| Running (Subject #9) | $42.13_{\pm 3.4}$ | $32.38_{\pm 3.9}$ | $20.78_{\pm 1.5}$ | $29.83_{\pm 1.4}$ | $18.5_{\pm 0.7}$ |
|  | EMLP | RPP | PER | MLP | Data Augmentation | Ours |
| Walking (Subject #35) | $7.01_{\pm 0.46}$ | $6.99_{\pm 0.21}$ | $7.48_{\pm 0.39}$ | $6.80_{\pm 0.18}$ | $6.37_{\pm 0.04}$ | $6.04_{\pm 0.09}$ |
| Running (Subject #9) | $57.38_{\pm 8.39}$ | $34.18_{\pm 2.00}$ | $33.03_{\pm 0.37}$ | $39.56_{\pm 2.25}$ | $40.23_{\pm 0.94}$ | $32.57_{\pm 1.47}$ |

We further illustrate a comparison on a real-world task, the Motion Capture dataset from CMU (2003). This dataset features 3D trajectory data that records a range of human motions, and the task involves predicting the final trajectory based on initial positions and velocities. We have reported results for two types of motion: Walking (Subject #35) and Running (Subject #9). We adhered to the standard experimental setup found in the literature (Han et al., 2022; Huang et al., 2022; Xu et al., 2024), employing a train/validation/test split of 200/600/600 for Walking and 200/240/240 for Running. Additional details can be found in Appendix B.2.

We apply our training procedure REMUL to the Transformer architecture and compare it with SE(3)-Transformer, Geometric Algebra Transformer (GATr), standard Transformer, and Transformer trained with data augmentation. We also compare with Equivariant MLP (Finzi et al., 2021b), Residual Pathway Priors (Finzi et al., 2021a), and Projection-Based Equivariance Regularizer (Kim et al., 2023a). As these architectures are designed specifically on MLP and linear layers, we apply our method to a standard MLP with a similar number of parameters.. Our results are presented in Table 1. For REMUL, we also provide plots on how the performance and equivariance error change *w.r.t.* the penalty parameter $\beta$ in Figure 3.

Table 1 indicates that when processing 3D positions related to human motions, both SE(3)-Transformer and GATr perform worse than the standard Transformer. This outcome is noteworthy because human motion inherently lacks symmetry along the vertical or gravity axis. Consequently, the assumption of equivariance across all axes may not be beneficial or even detrimental. In contrast, a standard Transformer trained with REMUL has the best performance in both tasks. Following Figure 3, there is a noticeable trade-off in model performance with different values of penalty parameter $\beta$. Best performance is observed at an intermediate level of equivariance, where the model balances between being too rigid (fully equivariant) and too flexible (non-equivariant). This finding underscores the importance of carefully considering the specific characteristics of the data and the task when designing equivariant architectures.

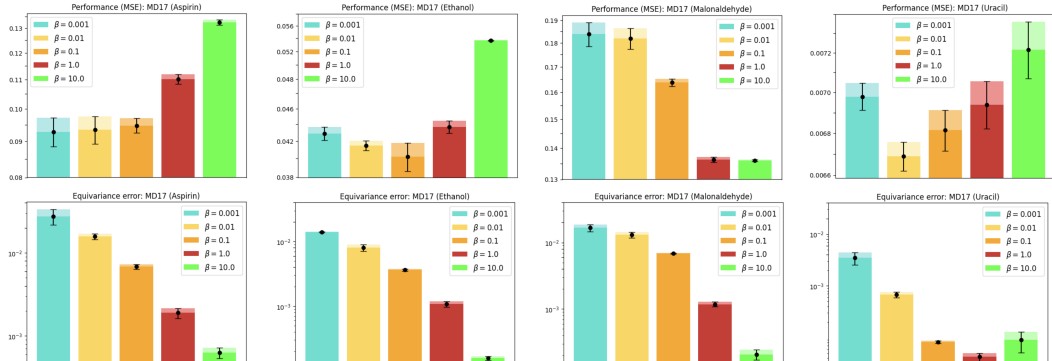

Figure 4: MD17 dataset: GNN trained with REMUL. The first row represents model performance (MSE), and the second row shows equivariance error. Columns from left to right show Aspirin, Ethanol, Malonaldehyde, and Uracil, respectively. The equivariance metric shown in this Figure is defined in Equation 12; we include the same plots for the second metric (Equation 13) in Appendix C.4 (Figure 10). The equivariance error decreases on all molecules with a higher value of $\beta$. In contrast, the required equivariance for best model performance varies for each molecule.

Table 2: Performance on MD17 dataset: MSE ($\times 10^{-2}$). REMUL procedure and data augmentation were applied to GNN. First, Second (highlighted). REMUL comes the best on six molecules and the second on two molecules.

|  | Aspirin | Benzene | Ethanol | Malonaldehyde | Naphthalene | Salicylic | Toluene | Uracil |
|---|---|---|---|---|---|---|---|---|
| EGNN | $14.41_{\pm 0.15}$ | $62.40_{\pm 0.53}$ | $4.64_{\pm 0.01}$ | $\mathbf{13.64}_{\pm 0.01}$ | $\mathbf{0.47}_{\pm 0.02}$ | $\mathbf{1.02}_{\pm 0.02}$ | $11.78_{\pm 0.07}$ | $\mathbf{0.64}_{\pm 0.01}$ |
| GNN | $\mathbf{9.26}_{\pm 0.40}$ | $\mathbf{26.13}_{\pm 0.11}$ | $\mathbf{4.26}_{\pm 0.03}$ | $18.45_{\pm 0.54}$ | $0.54_{\pm 0.001}$ | $\mathbf{1.02}_{\pm 0.02}$ | $\mathbf{9.93}_{\pm 0.82}$ | $0.70_{\pm 0.001}$ |
| Data Augmentation | $\mathbf{13.7}_{\pm 0.04}$ | $110.93_{\pm 5.3}$ | $5.74_{\pm 0.02}$ | $\mathbf{13.65}_{\pm 0.02}$ | $0.69_{\pm 0.001}$ | $1.33_{\pm 0.04}$ | $19.14_{\pm 0.001}$ | $0.73_{\pm 0.002}$ |
| REMUL | $\mathbf{9.28}_{\pm 0.40}$ | $\mathbf{25.95}_{\pm 0.18}$ | $\mathbf{4.02}_{\pm 0.16}$ | $\mathbf{13.59}_{\pm 0.03}$ | $\mathbf{0.54}_{\pm 0.001}$ | $\mathbf{0.99}_{\pm 0.001}$ | $\mathbf{9.38}_{\pm 0.20}$ | $\mathbf{0.67}_{\pm 0.001}$ |

## 6.3 MOLECULAR DYNAMICS

We also present a comparative analysis between constrained equivariant models and unconstrained models focusing on molecular dynamics, specifically predicting 3D molecule structures. We utilize the MD17 dataset (Chmiela et al., 2017), which comprises trajectories of eight small molecules. We use the same dataset split in Huang et al. (2022); Xu et al. (2024), allocating 500 samples for train, 2000 for validation, and 2000 for test. For this task, we selected the Equivariant Graph Neural Network (EGNN) architecture and its non-equivariant GNN counterpart, as presented in Satorras et al. (2021). We then apply REMUL procedure as well as data augmentation to the GNN architecture. Both architectures have the same hyperparameters. More information is indicated in Appendix B.3.

Our results are provided in Table 2. We illustrate how the performance and equivariance error of a GNN trained with REMUL vary across different molecules as a function of $\beta$, as shown in Figure 4 & Figure 9. From the results presented in Table 2, GNN trained with REMUL outperforms EGNN in six out of eight molecules. Interestingly, a standard GNN, without data augmentation or REMUL, surpasses the performance of EGNN for two molecules: Aspirin and Toluene. In Figure 4 & Figure 9, we observe that the optimal performance of each molecule is attained at different values of the penalty parameter $\beta$. For instance, Malonaldehyde exhibits a direct correlation between model performance and equivariance, where a higher $\beta$ yields better performance. Conversely, for most other molecules, there appears to be a pronounced trade-off where the best performance is achieved at a lower value of $\beta$. This is particularly evident with molecules like Aspirin, where a standard GNN architecture outperforms EGNN. We also plot the 3D structures of the eight molecules in Figure 11. Molecules such as Malonaldehyde, characterized by their symmetric components, might be ideally suited for equivariant design. However, this advantage does not apply to all molecules. Aspirin on the other side, might have an asymmetric structure and exhibit a range of interactions and dynamic states that equivariant models might simplify. Consequently, for such molecules, less equivariant models could potentially offer more accurate predictions.

## 6.4 COMPUTATIONAL COMPLEXITY

In this section, we report the computational time for the Geometric Algebra Transformer (GATr) and Transformer architectures. We selected models with an equivalent number of blocks and parameters for a fair comparison. Detailed configurations are provided in Appendix B.4. We measured the computational efficiency of each model by recording the time taken for both forward and backward passes during training, as well as inference time. In all comparisons, GATr architecture consistently required the highest time, being approximately ten times slower than Transformer architecture. Furthermore, GATr reached its memory capacity earlier, hitting an out-of-memory issue at a batch size of $2^{11}$. During inference, the computational speed for the Transformer trained with equivariance loss or data augmentation matches the standard Transformer, as all the differences applied in training. This results in an inference speed that is 10 times faster than GATr.

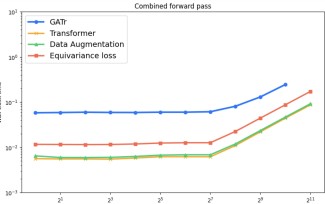 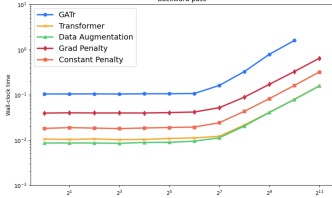 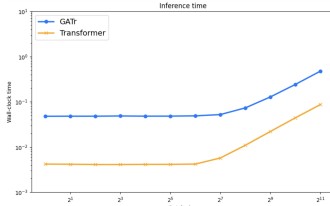

Figure 5: Computational time for Geometric Algebra Transformer (GATr) and Transformer architectures. Plots from left to right: Combined forward pass, backward pass, and inference time, respectively. GATr has the highest time in all scenarios.

## 6.5 LOSS SURFACE

In this section, we analyze the relative ease of training equivariant models compared to non-equivariant models by examining the loss surface around the achieved local minima for each model. We explore how each architecture influences the loss landscape when trained on the same task. However, due to the high dimensionality of parameter spaces in neural networks, visualizing their loss functions in three dimensions might be a significant challenge. We use the filter normalization method introduced by Li et al. (2018), which calculates the loss function along two randomly selected Gaussian directions in the parameters space, starting from the optimal parameters $\theta^*$ achieved at the end of training.

We visualize the loss surface of the Geometric Algebra Transformer (GATr) and Transformer in Figure 1, trained on the N-body dynamical system. We observe that the Transformer architecture exhibits a more favorable loss shape around its local minima, characterized by a convex structure. This might suggest that the optimization path for the Transformer is smoother and potentially easier to navigate during training, leading to more stable convergence. In contrast, the loss surface of GATr appears more erratic and rugged. This complexity in the loss landscape can indicate multiple local minima and a higher sensitivity to initial conditions or parameter settings. Such characteristics might complicate the training process, requiring more careful tuning of hyperparameters. We leave this for future work to analyze how the optimization path for each model behaves during training.

## 7 CONCLUSION

We introduced a novel, simple method for learning approximate equivariance in a non-constrained setting through optimization. We formulated equivariance as a new weighted loss that is simultaneously optimized with the objective loss during the training process. We demonstrated that we can control the level of approximate equivariance based on the specific requirements of the task. Our method competes with or outperforms constrained equivariant baselines, achieving up to 10 times faster inference speed and 2.5 times faster training speed.

**Limitations and Future Directions.** While we showed that unconstrained models exhibit a more convex loss landscape near the local minima compared to equivariant models, this observation is subject to certain limitations. Specifically, we did not account for the trajectories that these models traverse to reach their respective minima. Understanding the optimization paths and how different initialization settings influence these paths remains unexplored. In future work, we aim to analyze the optimization process of each model and how it behaves during training.

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

# A  ADDITIONAL TASK: JET FLOW BENCHMARK

We test our method on another real-world benchmark, the Jet Flow dataset used by Wang et al. (2022). The Jet Flow is a two-dimensional benchmark that captures turbulent velocity fields measured from NASA's multi-stream jet experiments. The dataset presents two primary tasks: Futruer: Given previous time steps of the flow field, the objective is to predict its future evolution. Domain: evaluate the model on different simulations from training. The dataset consists of $64 \times 23$ regions recorded from 24 stations.

We apply our method to Convolutional neural network (CNN) and compare it with E2CNN (Weiler & Cesa, 2019), and Relaxed Steerable Convolution (RSteer) (Wang et al., 2022). We follow the same training setup by Wang et al. (2022), which is summarized in Table 4 .

Table 3: Performance on Jet Flow dataset: RMSE. REMUL procedure is applied to standard CNN. First, Second (highlighted).

|  | Future | Domain |
| --- | --- | --- |
| E2CNN | $0.21_{\pm 0.02}$ | $0.27_{\pm 0.03}$ |
| RSteer | $\mathbf{0.17}_{\pm 0.01}$ | $\mathbf{0.16}_{\pm 0.01}$ |
| Ours | $\mathbf{0.16}_{\pm 0.003}$ | $\mathbf{0.18}_{\pm 0.003}$ |

Table 4: Hyperparameters settings for Jet Flow dataset.

| Hyperparameters | |
| --- | --- |
| #layers | 5 |
| #hidden dim | 16 |
| #kernel size | 3 |
| #epochs | 100 |
| #optimizer | Adam |
| #batch size | 16 |
| #lr | $1 \times 10^{-3}$ |

# B  IMPLEMENTATION DETAILS

## B.1  N-BODY DYNAMICAL SYSTEM

Following the methodology outlined in Brehmer et al. (2023), the dataset for the N-body system simulation encompasses four objects per sample. The center object is assigned a mass ranging from 1 to 10, whereas the other objects are uniformly positioned at a radius from 0.1 to 1.0 with masses between 0.01 and 0.1. We structured the datasets into two setups: in-distribution and out-of-distribution (OOD). Each sample in the in-distribution dataset is subjected to a random rotation within the range $[-10°, 10°]$. REMUL and data augmentation are trained with random rotations in the same range. Conversely, the OOD dataset is designed to evaluate the model's generalization capabilities by incorporating extreme rotational perturbations, specifically with angles set within the ranges $[-180°, -90°]$ and $[90°, 180°]$. We trained on 100 samples, and each of the validation, test, and OOD datasets contains 5000 samples. For models hyperparameters and training, we follow the same settings in Brehmer et al. (2023), summarized in Table 5. For REMUL, initial $\alpha = 1$.

## B.2  MOTION CAPTURE

Motion Capture dataset by CMU (2003) features 3D trajectory data that records a range of human motions, and the task involves predicting the final trajectory based on initial positions and velocities. We have reported results for two types of motion: Walking (Subject #35) and Running (Subject #9).

Table 5: Hyperparameters settings for N-body dynamical system.

| Hyperparameters | Geometric Algebra Transformer | SE(3)-Transformer | Transformer |
|---|---|---|---|
| #attention blocks | 10 | - | 10 |
| #channels | 128 | 8 | 384 |
| #attention heads | 8 | 1 | 8 |
| #multivector | 16 | - | - |
| #layers | - | 4 | - |
| #degrees | - | 4 | - |
| #training steps | 50000 | 50000 | 50000 |
| #optimizer | Adam | Adam | Adam |
| #batch size | 64 | 64 | 64 |
| #lr | $3 \times 10^{-4}$ | $3 \times 10^{-4}$ | $3 \times 10^{-4}$ |

Following the standard experimental setup in the literature on this task (Han et al., 2022; Huang et al., 2022; Xu et al., 2024), we apply a train/validation/test split of 200/600/600 for Walking and 200/240/240 for Running. The interval between trajectories, $\Delta T = 30$ for both tasks. For model hyperparameters, we fine-tuned around the same in Table 5 and found it the best for each model except for the Geometric Algebra Transformer we increased the attention blocks to 12. We train each model for 2000 epochs with batch size = 12. For the MLP comparison, all the models and baselines have the same number of layers and parameters. (details in Table 6).

Table 6: Hyperparameters settings for Motion Capture dataset.

| Hyperparameters | Geometric Algebra Transformer | SE(3)-Transformer | Transformer |
|---|---|---|---|
| #attention blocks | 12 | - | 10 |
| #channels | 128 | 8 | 384 |
| #attention heads | 8 | 1 | 8 |
| #multivector | 16 | - | - |
| #layers | - | 4 | - |
| #degrees | - | 4 | - |
| #epochs | 2000 | 2000 | 2000 |
| #optimizer | Adam | Adam | Adam |
| #batch size | 12 | 12 | 12 |
| #lr | $3 \times 10^{-4}$ | $3 \times 10^{-4}$ | $3 \times 10^{-4}$ |

| Hyperparameters | Equivariant MLP | RPP | PER | standard MLP |
|---|---|---|---|---|
| #hidden dim | 532 | 348 | 532 | 680 |
| #layers | 3 | 3 | 3 | 3 |

### B.3 MOLECULAR DYNAMICS

MD17 dataset (Chmiela et al., 2017) is a molecular dynamics benchmark that contains the trajectories of eight small molecules (Aspirin, Benzene, Ethanol, Malonaldehyde Naphthalene, Salicylic, Toluene, Uraci). We use the same dataset split in Huang et al. (2022); Xu et al. (2024), allocating 500 samples for train, 2000 for validation, and 2000 for test. The interval between trajectories, $\Delta T = 5000$. We selected the Equivariant Graph Neural Networks (EGNN) architecture and its non-equivariant version GNN, as introduced by Satorras et al. (2021). The input for GNN architecture is the initial positions along with atom types. Both architectures have the same hyperparameters, details in Table 7. For REMUL, $\alpha = 1$.

### B.4 COMPUTATIONAL COMPLEXITY

In the computational experiment of Geometric Algebra Transformer (GATr) and Transformer, we selected models with an equivalent number of blocks and parameters. GATr incorporates a unique

Table 7: Hyperparameters settings for MD17 dataset.

| Hyperparameters | |
|---|---|
| #layers | 4 |
| #hidden dim | 64 |
| #epochs | 500 |
| #optimizer | Adam |
| #batch size | 200 |
| #lr | $5 \times 10^{-4}$ |

design that includes a multivector parameter; we adjusted the Transformer architecture to match the parameter count of GATr. Both models have around 2.6M parameters, detailed configurations are provided in Table 8. SE(3)-Transformer gives out of memory for this setting. We selected a uniformly random Gaussian input with 20 nodes and 7 features dimension. We measured the computational efficiency of each model by recording the time taken for both forward and backward passes during training, as well as the inference time as a function of batch size. For each value, we took the average over 10 runs with Nvidia A10 GPU.

Table 8: Hyperparameters settings for Computational Complexity.

| Hyperparameters | Geometric Algebra Transformer | Transformer |
|---|---|---|
| #attention blocks | 12 | 12 |
| #channels | 128 | 168 |
| #attention heads | 8 | 8 |
| #multivector | 16 | - |

## C ADDITIONAL EXPERIMENTS

In this section, we include additional results on the three tasks (N-Body Dynamical System, Motion Capture, and Molecular Dynamics), using the equivariance measure defined in (Equation 13) which is consistent with our results in the paper. We also include molecules from the MD17 dataset, along with visualizations of their structures in both 2D and 3D.

### C.1 N-BODY DYNAMICAL SYSTEM

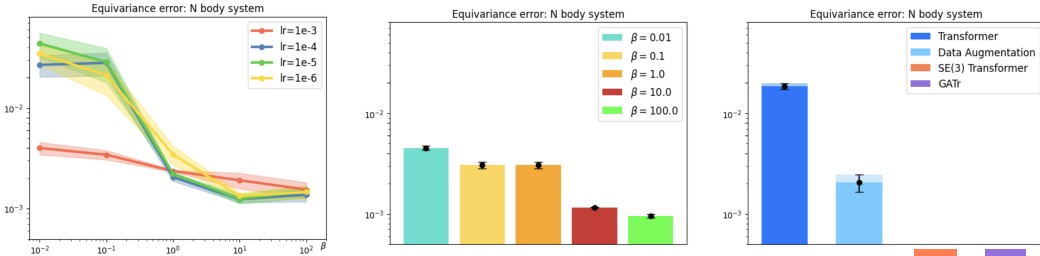

Figure 6: N-body dynamical system. The second equivariance measure (defined in Equation 13). Plots from left to right: The first shows the Transformer trained with REMUL (gradual penalty), the second with a constant penalty, and the third presents the baselines (equivariant models, standard Transformer, and data augmentation). SE(3)-Transformer and GATr have a small equivariance error below the range of the plots ($3.1e^{-9}$ and $1.22e^{-14}$ respectively).

## C.2 NUMBER OF GROUP SAMPLES

In this section, we conduct ablation studies on the number of samples required from the symmetry group during training. We compare our training procedure, REMUL, with data augmentation method. We follow the same training details and hyperparameters indicated in Appendix B.1. As shown in Figure 7, REMUL achieves better performance using fewer samples from the symmetry group compared to data augmentation.

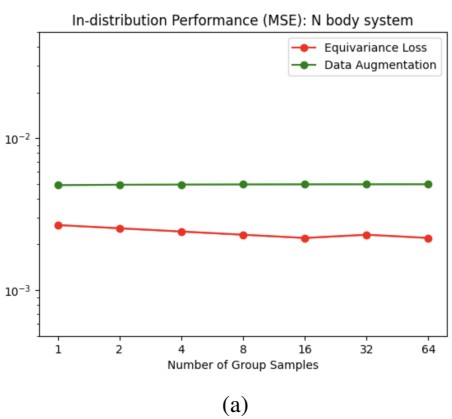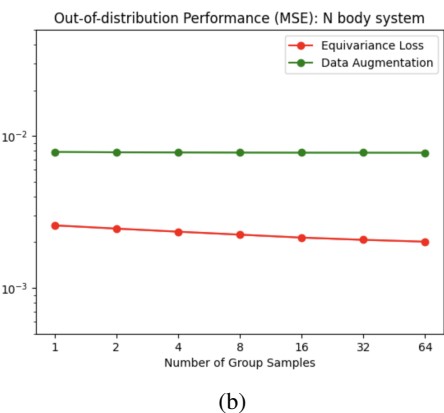

(a)                                         (b)

Figure 7: Motion Capture dataset: Transformer trained with REMUL. The second equivariance measure (defined in Equation 13). Left: Walking task (Subject #35) and right: Running task (Subject #9).

## C.3 MOTION CAPTURE

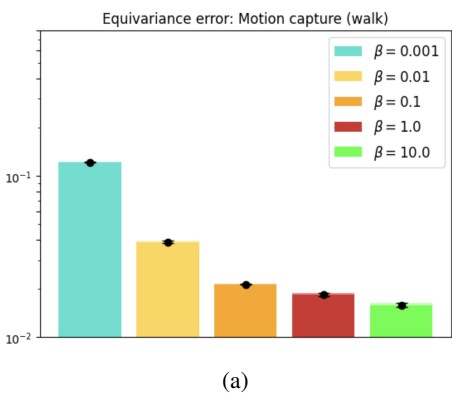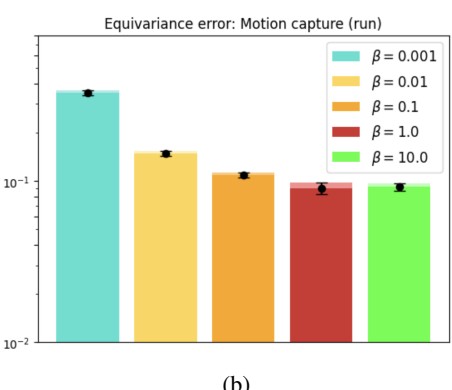

(a)                                         (b)

Figure 8: Motion Capture dataset: Transformer trained with REMUL. The second equivariance measure (defined in Equation 13). Left: Walking task (Subject #35) and right: Running task (Subject #9).

## C.4 MOLECULAR DYNAMICS

Figure 9: MD17 dataset: GNN trained with REMUL. The first column is model performance (MSE), and the second column is equivariance error (Equation 12). Rows from top to bottom represent Benzene, Naphthalene, Salicylic, and Toluene, respectively.

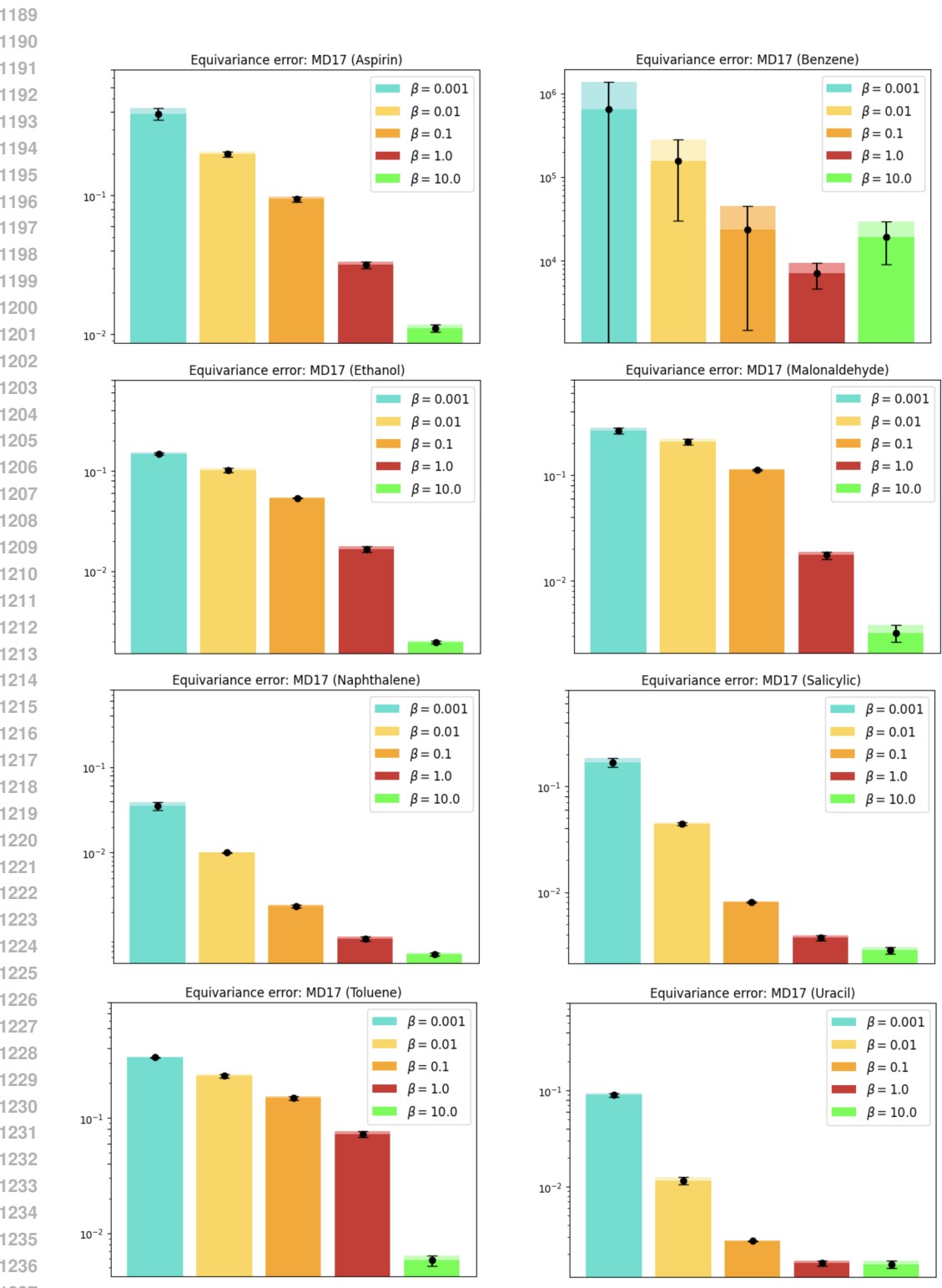

Figure 10: MD17 dataset: GNN trained with REMUL. The second equivariance measure (defined in Equation 13).

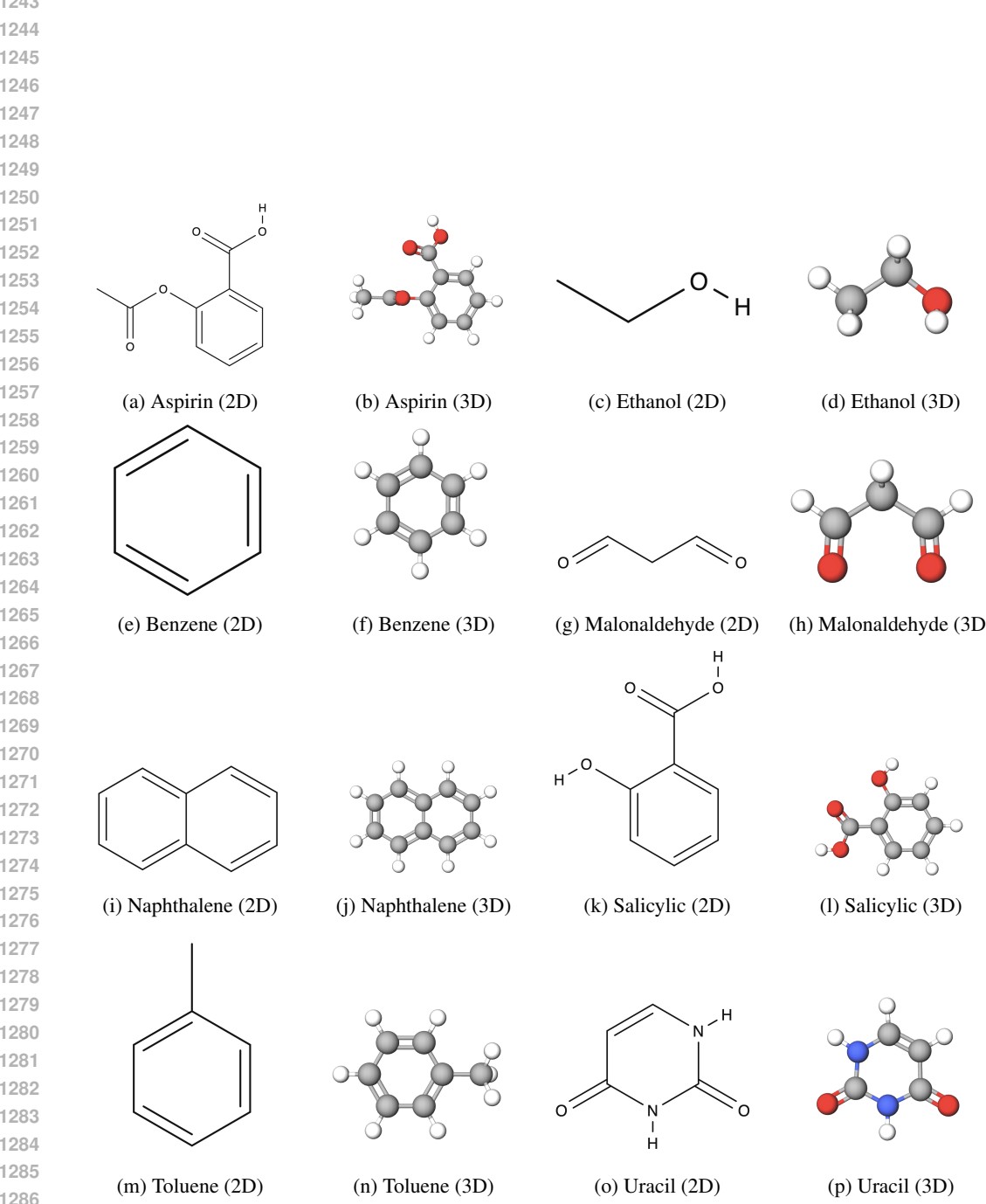

Figure 11: MD17 molecules structures.

