# OpenReview forum: "Learning Symmetries through Loss Landscape"
_ICLR.cc/2025/Conference — Submitted to ICLR 2025_

### Official Review · Reviewer_JQbk · 2024-10-28

**Soundness:** 1
**Presentation:** 3
**Contribution:** 1
**Rating:** 5
**Confidence:** 4

**Summary:**

In this paper, the authors analyze the application of unconstrained models in equivariant tasks by conducting a comprehensive analysis of unconstrained models, comparing their performance and computational efficiency against equivariant models. Besided, the authors introduce a novel, simple loss function that enables these models to approximate symmetries, which can be optimized during training.

**Strengths:**

Relaxing equivariance is a valuable research direction that can break through the constraints on generalization or expressive power caused by strictly equivariant operations.

**Weaknesses:**

1. The primary concern is the authors' motivation. The idea of using group transformations for data augmentation is native, but for many equivariant tasks, it is challenging to obtain a general model through data sampling due to the bias introduced by limited sampling. For instance, for point clouds or molecules, sampling across all angles would expand the dataset by hundreds of times and still struggle to enable the model to effectively learn fine-grained rotation equivariance. I suggest the authors validate their approach on common 3D datasets such as QM9 or ModelNet40.

2. The authors base their introduction in the first three sections on general equivariance, yet the impact of different equivariance groups on algorithms varies. For example, permutation equivariance and translation equivariance can be directly covered by simple operations, making the paper's method inapplicable. The authors should specify which equivariant tasks their method focus on.

3. Relaxing equivariance is a widely discussed topic, and the authors lack relevant citations and analysis [1] [2] [3] [4]. Moreover, the main advantage of unconstrained models is their ability to learn more complex features. It is worth noting that strictly equivariant operations can limit the expressive power of GNNs [5] [6], but unconstrained models may surpass these limitations. Additionally, some tasks are not strictly equivariant, allowing unconstrained models to be applicable. The authors' emphasis on the lower computational complexity of unconstrained operations is incorrect. In the 3D domain, models like torchmd are strictly equivariant yet have low complexity.

    [1] Residual pathway priors for soft equivariance constraints, Finzi, et al.

    [2] Approximately equivariant networks for imperfectly symmetric dynamics. Wang, et al.

    [3] Relaxing equivariance constraints with non-stationary continuous filters. van der Ouderaa, et al.

    [4] Learning Partial Equivariances from Data. Romero, et al.

    [5] On the Universality of Rotation Equivariant Point Cloud Networks. Nadav Dym, Haggai Maron.

    [6] On the Expressive Power of Geometric Graph Neural Networks. Chaitanya K. Joshi, Cristian Bodnar, Simon V. Mathis, Taco Cohen, Pietro Liò.

4. I do not understand how the loss surface in Figure 1 was created and why it demonstrates the advantages of unconstrained models.

5. There are numerous issues with the paper's presentation:

    (a) The equations in lines 218, 224, and 227 lack numbering.

    (b) In line 215, the definition of G is finite, which is problematic for integrals where the group size can be infinite. Most groups mentioned in the paper are infinite, and I do not understand why the authors restrict groups to being finite in their initial definition.

    (c) All the references have formatting issues because none of them specify the source of the papers. For instance, "Equivariant Graph Hierarchy-Based Neural Networks" in your paper is accepted in NeurIPS 2022, not arxiv.

    (d) Appendix B is incomplete; several titles are clustered together without any explanatory text.

**Questions:**

See weekness.

---

> ### Author Response · Authors · 2024-11-22
>
> We thank the reviewer for their time and all the comments, which have helped us improve our paper and indicate our contribution. We are also open to answer any further questions.
>
> **Limited group transformations:**
>
> - We would like to highlight to the reviewer that our experiments primarily focus on point clouds and molecules across three key tasks: 3D Dynamical Systems, Motion Capture, and Molecular Dynamics. While we acknowledge that fully equivariant tasks, such as rotation, can be computationally intensive when sampling across all angles, our empirical results show that applying a single random rotation during training is sufficient to achieve performance comparable to equivariant baselines.
>
> - We also conduct additional experiments with different numbers of samples from the symmetry group during training comparing our method and data augmentation. Our new results confirm that we can achieve reasonable performance using fewer samples from the symmetry group. We included the results in Appendix C.2 in the updated version.
>
> **Symmetry group:**
>
> - We thank the reviewer for highlighting the need to specify the equivariant tasks our method focuses on. In this work, we consider SE(3) symmetry group (i.e. group of rotations and translations).  We have clarified this in the Introduction section of the updated manuscript.
>
>
> **Relaxed equivariance and unconstrained models**
>
> - We have updated our manuscript to consider the relevant works on relaxed equivariance in the Related Work section.
>
> - We have added new results on the Motion Capture task. We compare our method against Residual Pathway Priors (RPP) [1], Projection-Based Equivariance Regularizer (PER) [2], and Equivariant MLP (EMLP) [3]. As these architectures are designed based on linear layers and MLP, we apply the augmented loss to standard MLP with a similar number of layers and parameters.
>
> - We also include a new task on the Jet Flow dataset used by [4] (a two-dimensional benchmark that captures turbulent velocity fields): We apply our method to a Convolutional neural network (CNN) and compare it with Relaxed Steerable Convolution (RSteer) [4] and E2CNN [5] (more details in Appendix A in the paper).
>
> - We clarified in the updated version that the high computational cost comes particularly with those relying on spherical harmonics and irreducible representations (Introduction Section).
>
> -  We thank the reviewer for the point on the limited expressive power and included it in the updated version. We also added some examples of tasks that exhibit approximate equivariance (Introduction Section).
>
>
> 1. Residual Pathway Priors for Soft Equivariance Constraints. Finzi et al., NeurIPS 2021.
>
> 2. Regularizing Towards Soft Equivariance Under Mixed Symmetries.  Kim et al., ICML 2023.
>
> 3. A Practical Method for Constructing Equivariant Multilayer Perceptrons for Arbitrary Matrix Groups. Finzi et al., 2021.
>
> 4. Approximately Equivariant Networks for Imperfectly Symmetric Dynamics. Wang et al., ICML 2022.
>
> 5. General E(2)-Equivariant Steerable CNNs. Weiler et al., NeurIPS 2019.

---

> > ### Author Response · Authors · 2024-11-22
> >
> > **Loss surface:**
> >
> > Due to the high dimensionality of parameter spaces in neural networks, visualizing the
> > loss functions in three dimensions might be a significant challenge. Some works studied this through 1D or 2D interpolations [1, 2]. Among these, we use the filter normalization method [3], which calculates the loss function along two randomly selected Gaussian directions in the parameters space, starting from the optimal parameters achieved at the end of training (local minima).
> > Having a flat or smooth surface around the minima is associated with better generalization and easier training [3, 4, 5]. However, we agree with the reviewer this still requires further study, and we plan to explore this direction in future work.
> >
> > 1. Qualitatively characterizing neural network optimization problems. ICLR, 2015.
> >
> > 2. An empirical analysis of the optimization of deep network loss surfaces.
> >
> > 3. Visualizing the Loss Landscape of Neural Nets. Li et al,  NeurIPS, 2018.
> >
> >
> > 4. Understanding the Difficulty of Training Transformers
> >
> > 5. Entropy-SGD: Biasing Gradient Descent Into Wide Valleys. Chaudhari1 et al., ICLR, 2017.
> >
> > **Other issues:**
> >
> > - We thank the reviewer for pointing these out. We have updated the paper incorporating all the comments, including the missing equation numbers, corrected references and formatting, and an explanation of the Appendix.
> >
> > - We agree with the reviewer that we consider infinite groups in our work, we modified the definition in Section 4 and explained that we approximate the integral with the Monte Carlo approach.
> >
> > We sincerely hope that we have addressed the concerns of the reviewer satisfactorily in the revised version and would kindly ask the reviewer to update their score accordingly.

---

> ### Comment · Reviewer_JQbk · 2024-11-25
> **Official Comment by Reviewer JQbk**
>
> Thank you for your response. I am pleased that you have carefully read my questions and taken my suggestions. Now, it seems that the paper has indeed improved. However, I still cannot change my score to a positive opinion for several reasons:
>
> 1. We often consider equivariance when focus on some complex systems. In this case, adding a random rotation to the loss once is not enough. The model might be able to witness all possible SO(3) transformations by increasing the number of epochs, but this would also greatly increase the training cost. I recommend that the authors test their algorithms on some complex systems, such as QM9, GEOM-Drug, OC20, which are more concerned 3D systems in the field of computational chemistry.
>
> 2. Equivariant models are not actually expensive. For example, the TorchMD-Net model only uses low-degree equivariance $l=1$, and it can achieve good results on both QM9 and MD17. Some equivariant models are expensive since the tensor product on higher-degree spherical harmonic representations is very time-consuming.
>
> 3. The comparative experiments in the paper still contain irrationalities: MD17 includes both energy and forces properties data, but I am not sure which one the authors are comparing in Table 2 (if the authors have stated this, I apologize). Why not compare both energy and forces? Additionally, the comparison in this table is also unreasonable: why is the error for EGNN so large (MSE for Benzene is 62.40)? The MD17 dataset is a simple molecular dataset, and it should not perform so poorly. You can look at models like TorchMD-Net, Equiformer, etc., where the results on MD17 are almost saturated. Furthermore, if the authors want to prove that your loss helps reduce the computational cost of equivariant models, you should list some detail results of computational metrics (In Figure 5, we only observe that your method outperforms GATr).
>
> In conclusion, I am sorry that I cannot change my score because I believe this paper does not address the key issues of 3D equivariance (performance and computational cost). Adding a random rotation to the loss is a natural idea, and perhaps the authors can look into whether there are similar works that have tried this approach. However, I acknowledge the author's research direction. I believe that approximate or learnable equivariance is valuable because strict equivariance can lead to a loss of expressive power, and unconstrained neural networks are more likely to learn some complex 3D features. The authors could also design algorithms from the perspective of reducing the training cost of equivariant models. Perhaps modifying the loss could solve the bottleneck in equivariance. I encourage the authors to delve deeper into their research.

---

> ### Author Response · Authors · 2024-11-26
>
> We thank the reviewer for their time and for engaging with us in the discussion. We appreciate the reviewer’s acknowledgment
> that we have taken their suggestions and the improvement of the paper.
>
> - **"We often consider equivariance when focus on some complex systems. In this case, adding a random rotation to the loss once is not enough".** As we pointed out before, we conduct additional experiments with different numbers of samples from the $SO(3)$ group during training comparing our method and data augmentation. In practice, our new results confirm that we can achieve reasonable performance using fewer samples from the symmetry group. Please see the results in Appendix C.2 in the updated version.
>
> - **"The model might be able to witness all possible SO(3) transformations by increasing the number of epochs, but this would also greatly increase the
> training cost."** We would like to point out to the reviewer that we trained all the models, unconstrained and equivariant models, in all the experiments and benchmarks using the same number of epochs. Please see the details in Appendix B.
> - **"I recommend that the authors test their algorithms on some complex systems, such as QM9, GEOM-Drug, OC20, which are more concerned 3D systems in the field of computational chemistry."** We want to clarify to the reviewer that both Motion Capture and Molecular Dynamics (MD17) are widely used 3D benchmarks in the literature. Please see for example [1, 2, 3, 4, 5, 6, 7, 8, 9, 10].
>
> - **"Equivariant models are not actually expensive. For example, the TorchMD-Net model only uses low-degree equivariance, and it can achieve good results on both QM9 and MD17".** We thank the reviewer for pointing out the example of TorchMD-Net. However, there is no contradiction between that and what we pointed out in our paper.  Methods that use spherical harmonics and higher-order tensor products (which are more expressive) are computationally expensive. Using a lower degree could reduce the computational cost but it comes at the trade-off of performance. Please see for example [11, 12, 13].
>
> - **"Some equivariant models are expensive since the tensor product on higher-degree spherical harmonic representations is very time-consuming".** We appropriate the reviewer's acknowledgment that 'tensor product on higher-degree spherical harmonic representations is very time-consuming'. However, this is what we indicated in our paper. Please see the introduction section.
>
> - **"The comparative experiments in the paper still contain irrationalities: MD17 includes both energy and forces properties data, but I am not sure which one the authors are comparing in Table 2 (if the authors have stated this, I apologize). Why not compare both energy and forces?"** We would like to clarify to the reviewer that  MD17 benchmark has two common tasks in the literature:
>     - Invariant Task: Predicting energies given molecular states/ positions (Please see for eg [1, 2, 3, 4]).
>     - Equivariant Task: Predicting molecular states/ positions after a specific number of time steps given initial states/ positions (Please see for eg [5, 6, 7, 8]).
>
>
> In this work, we focus on the equivariant task, following the same previous work on this task.  Our primary objective is to compare unconstrained models with their corresponding equivariant versions (GNN vs EGNN).
>
> - **"Additionally, the comparison in this table is also unreasonable: why is the error for EGNN so large (MSE for Benzene is 62.40)? The MD17 dataset is a simple molecular dataset, and it should not perform so poorly. You can look at models like TorchMD-Net, Equiformer".** We would like to point out to the reviewer that we follow the literature on the equivariant task where the models they suggested focus on the invariant task.
> However, we are not the first to report the results for EGNN on this task and we follow the same setup in the literature. Please see for example [5, 6, 7, 8]. Additionally, all the training details are reported in Appendix B.3 and we also plan to publish the full code if it has been accepted.
> In our work, we observe that the optimal performance of each molecule is attained at different values of the
> penalty parameter $\beta$. For instance, Malonaldehyde exhibits a direct correlation between model performance and equivariance, where a higher penalty parameter $\beta$ yields better performance. For this, EGNN achieved good performance. Conversely, for some molecules, there is a trade-off where the best performance is achieved
> at a lower value of $\beta$. Please see Section 6.3 in the paper.

---

> ### Author Response · Authors · 2024-11-26
>
> - **"Furthermore, if the authors want to prove that your loss helps reduce the computational cost of equivariant models, you should list some detail results of computational metrics (In Figure 5, we only observe that your method outperforms GATr)"** We thank the reviewer for this point. In Figure 5, we report the wall-clock time for the Geometric Algebra Transformer (GATr) and
> Transformer architectures. We selected models with an equivalent number of blocks and parameters for a fair comparison (please see Appendix B for details).
> We measured the computational efficiency of each model by recording the time taken for both forward and backward passes during training, as well as inference time. We show that Transformer can achieve up to $10$ times
> faster inference speed and $2.5$ times faster training speed compared to GATr.
> However, although we have included the GATr architecture in our discussion, GATr itself is faster than many equivariant architectures, such as SEGNN and SE(3)-Transformer (please see [14]).
>
>
> - **"Adding a random rotation to the loss is a natural idea, and perhaps the authors can look into whether there are similar works that have tried this approach."** We thank the reviewer for this point. However, we already included a related work section with a detailed discussion of the existing literature on approximate equivariance, with all the differences between these approaches and our proposed method.
> If there are any further questions or specific aspects that we have not addressed, or the reviewer would like to point out, we are happy to provide additional clarifications.
>
> - **"The authors could also design algorithms from the perspective of reducing the training cost of equivariant models. Perhaps modifying the loss could solve the bottleneck in equivariance. I encourage the authors to delve deeper into their research."** We thank the reviewer for their suggestions. However, we believe that both directions are valuable and have a potential impact. In this work, we focus on comparing existing equivariant models with their unconstrained counterparts across a diverse set of benchmarks. Specifically, we evaluate Transformers, Graph Neural Networks (GNNs), Convolutional Neural Networks (CNNs), and Multi-Layer Perceptrons (MLPs) on four distinct tasks: Dynamical Systems, Motion Capture, Molecular Dynamics, and Jet Flow benchmarks. We aim to provide valuable insights into the performance, scalability, and applicability of unconstrained vs equivariant models across various domains.
>
> 1. SchNet: A continuous-filter convolutional neural network for modeling quantum interactions. Schütt et al., NeurIPS 2017.
>
> 2. Rotation Invariant Graph Neural Networks using Spin Convolutions. Shuaibi1 et al.,  2021.
>
> 3. Spherical Message Passing for 3D Graph Networks. Liu et al., ICLR 2022.
>
> 4. Symmetry-Informed Geometric Representation for Molecules, Proteins, and Crystalline Materials. Liu et al., NeurIPS 2023.
>
> 5. Equivariant graph mechanics networks with constraints. Huang et al., ICLR 2022.
>
> 6. EqMotion: Equivariant Multi-Agent Motion Prediction With Invariant Interaction Reasoning. Xu et al., CVPR 2023.
>
> 7. Equivariant Spatio-Temporal Attentive Graph Networks to Simulate Physical Dynamics. Wu et al., NeurIPS 2023.
>
> 8. Equivariant Graph Neural Operator for Modeling 3D Dynamics. Xu et al., ICML 2024.
>
> 9. Equivariant Graph Hierarchy-Based Neural Networks. Han  et al., NeurIPS 20222.
>
> 10. Clifford Group Equivariant Simplicial Message Passing Networks. Liu et al., ICLR 2024.
>
> 11. Equiformer: Equivariant Graph Attention Transformer for 3D Atomistic Graphs Liao et al., ICLR 2023.
>
> 12. EquiformerV2: Improved Equivariant Transformer for Scaling to Higher-Degree Representations Liao et al., ICLR 2024.
>
> 13. Enabling Efficient Equivariant Operations in the Fourier Basis via Gaunt Tensor Products. Luo et al., ICLR 2024.
>
> 14. Geometric Algebra Transformer. Brehmer et al., NeurIPS 2023.

---

> > ### Comment · Reviewer_JQbk · 2024-11-27
> >
> > Thank you for your patient responses; your responses have indeed addressed some of my concerns, especially on the experimental results of MD17. I am willing to raise my score to 5.
> >
> > I have briefly reviewed the molecular dynamics articles you provided, and I apologize for my lack of professionalism in this field. In fact, my understanding of MD17 has always been focused on force and energy prediction, which is not comprehensive. I am willing to lower my confidence to 4.
> >
> > Finally, the rebuttal period is coming to an end, and I won't have the chance to hear back from you, but I will take the time to read the references you've listed and continue to consider whether to change the score.

---

> > > ### Author Response · Authors · 2024-12-02
> > > **Follow Up**
> > >
> > > Dear reviewer,
> > >
> > > Thank you once again for your time and the valuable feedback you have provided. We did our best to answer your questions and follow your comments in the revised version.
> > > As the discussion period is ending soon, we kindly ask if the reviewer have made a decision on whether to raise their score, as described in their latest response, or if they have any further questions.

---

### Official Review · Reviewer_EQn6 · 2024-11-03

**Soundness:** 2
**Presentation:** 3
**Contribution:** 2
**Rating:** 6
**Confidence:** 4

**Summary:**

The paper attempts to build equivariance into unconstrained models by posing equivariance as a constrained optimization problem, which can, in turn, also control the level of approximate equivariance in the models. The authors demonstrate results in N-body dynamical systems, motion capture, and molecule dynamics, and they analyze the effect of the level of approximate equivariance on task performance.

**Strengths:**

- The paper is well-motivated and clearly written (particularly the sections on background and methods).
- The limitation section discusses an important limitation of the interplay between optimization paths and loss landscape.
- The experiments are conducted in different domains and examine several essential aspects of the algorithm, giving more insights into the method and how levels of equivariance can affect downstream task performance.

**Weaknesses:**

- **Related work**:
   - Although the paper uses equivariance as a constrained optimization problem and discusses it in the context of unconstrained models, it misses several crucial relevant works. Discussion of these works would help to place the submission in the literature and give a view of how this work differs from and compares to existing works.
  - Learning equivariance from data [1, 2], approximate/soft equivariance [3, 4, 5, 6], equivariance as a constrained optimization problem [7, 8], and equivariance with unconstrained models [9, 10, 11, 12].
  - Can the authors highlight the differences from Sec 3.1 and Sec. 3.2 of [10]?

- **$\beta$ and $\alpha$ as hyperparameters**:
  - The authors suggest that the level or extent of equivariance can be controlled with $\beta$ and $\alpha$ - is there a formal way to define this "level" of equivariance or is it an intuition tied to the loss itself, i.e., higher $\frac{\beta}{\alpha}$ indicates more equivariant?
  - Next, how would someone know the optimal level of equivariance while using your proposed algorithm - $\beta$ is not learned, and the results indicate that the optimal $\beta$ can be identified from the test data results, which is not ideal. Rephrasing this, how do you know how much equivariance is required for the task, and thus what values of $\alpha$ and $\beta$ to set?


- **Methodology**:
  - How will your algorithm work if group $G$ is unknown?
  - How can your method reasonably approximate equivariance if $G$ is very large and the duration of training is not enough?
  - The highest level of equivariance is when $\alpha = 0$ and $\beta=1$. However, this is equivalent to data augmentation, which does not guarantee exact equivariance. Can your algorithm guarantee exact equivariance?
  - While the trends are consistent for both metrics, as reported in the paper, it might be helpful to discuss which metric - Eq. 9 or Eq. 10 is better suited for evaluation. How does Equation 9 work (or make sense) when $f(x)$ is non-scalar?
  - For Motion Capture, if the symmetry constraints are already known, instead of complete SE(3) equivariant baselines, why didn't the authors select appropriate equivariant models that are equivariant to the required SE(3) subgroup or consider symmetry breaking [14, 15]? What $G$ did your algorithm use? If it is the subgroup of SE(3), then it is an unfair comparison.

- **Minor spelling errors**:
  - L156 "requiring equivariant into" should be "requiring equivariance in"
  - L396 "it is" should be "its"


**References**:
1. Equivariance Discovery by Learned Parameter-Sharing. Yeh et al., AISTATS 2022.
2. Learning Equivariances and Partial Equivariances from Data. Romero et al., NeurIPS 2022.
3. Learning Layer-wise Equivariances Automatically using Gradients. Ouderaa et al., 2023.
4. Residual Pathway Priors for Soft Equivariance Constraints. Finzi et al., NeurIPS 2021.
5. Almost Equivariance via Lie Algebra Convolutions. McNeela et al., 2024.
6. Regularizing Towards Soft Equivariance Under Mixed Symmetries. Kim et al., ICML 2023.
7. Improved Canonicalization for Model Agnostic Equivariance. Panigrahi et al., 2024.
8. Structuring Representations Using Group Invariants. Shakerinava et al., NeurIPS 2022.
9. Equivariance with Learned Canonicalization Functions. Kaba et al., ICML 2023.
10. Equivariant adaptation of large pretrained models. Mondal et al., NeurIPS 2023.
11. Equi-Tuning: Group Equivariant Fine-Tuning of Pretrained Models. Basu et al., AAAI 2023.
12. Learning Probabilistic Symmetrization for Architecture Agnostic Equivariance. Kim et al., NeurIPS 2023.
13. Steerable Equivariant Representation Learning. Bhardwaj et al., 2023
14. Symmetry breaking and equivariant neural networks. Kaba et al., NeurIPS NeuReps workshop 2023
15. Improving Equivariant Networks with Probabilistic Symmetry Breaking. Lawrence et al., ICML GRaM workshop 2024.

**Questions:**

- How will your algorithm fare when there are data constraints? Equivariant models are inherently data efficient, but your algorithm does not seem to be.
- The loss landscape plots depend on the selected directions - so how can we infer from just two random directions that the loss landscape is better for Transformers or GATr? The optimization paths should affect, and although it is discussed to some extent in limitations, it would be better if there is more discussion on this.

Most of the other questions I had are listed in the Weaknesses section. I will be happy to improve the score if the authors address the questions and weaknesses with supportive evidence during the discussion phase.

---

> ### Author Response · Authors · 2024-11-22
>
> We thank the reviewer for their time and all the comments, which have helped us improve our paper and indicate our contribution. We are also open to answer any further questions.
>
> **Related Work:**
>
> - We thank the reviewer for these points.  We have added a new section discussing learning symmetries' approximate equivariance and included comparisons in the related work section.
>
> - We have included all the citations and comparisons pointed out by the reviewer.
>
> - The method in [1] used a canonicalization approach (i.e. additional trainable canonicalization network) to orient the input before the main network. For stable canonicalization, the authors used prior distribution over the symmetry group. However, in our work, we didn't use any additional canonicalization network, and we directly learned equivariance through the unconstrained model.
>
> 1. Equivariant adaptation of large pretrained models. Mondal et al., NeurIPS 2023.
>
> **$\alpha$ and $\beta$ as hyperparameters:**
>
> -  We agreed with the reviewer that higher $\frac{\beta}{\alpha}$ leads to more equivariant function. However, this will not indicate if the function can approximate equivariance or not, or to what extent, as this will be specific to the parameters $\alpha$ and $\beta$. In other words, we can define the relative equivariance as a fraction of $\beta$ and $\alpha$, but we also want to measure the level of approximate equivariance the model learns, for this we introduced the equivariance metric.
>
> - $\alpha$ and $\beta$ will be treated as additional parameters that are determined from the validation set. Similar to other parameters in the training procedure (eg learning rate, batch size, number of layers in architecture, etc).
>
> **Methodology:**
>
> **Symmetry group:** In this work we consider SE(3) symmetry group (i.e. the group of rotations and translations) and SE(3) equivariant architectures.  We have clarified this in the Introduction section of the updated manuscript. We are happy to extend this in future for unknown group (eg  Lie algebra convolutional by [1]).
>
> **if $G$ is very large**. As we focus on SE(3) symmetry group, this already considered a very large/ infinite group. We noticed that empirically our algorithm is sufficient to achieve competitive performance with equivariant baselines using very small number of samples. For example, in the N-Body dynamical system, we train all the models using $100$ samples.
>
> **The highest level of equivariance is when $\alpha = 0$ and $\beta = 1$**. It is important to note that $\beta$ is not limited to $1$. In our formulation, $\beta$ can take any positive real value. By increasing $\beta$ we can place greater emphasis on enforcing equivariance in the model and reduce the equivariance error (Figure 2 in the paper). We observed that this approach is empirically sufficient to achieve competitive performance with advanced equivariant architectures, such as the Geometric Algebra Transformer, when applied to fully equivariant tasks like the N-body dynamical system.
>
> **Which metric is better suited for evaluation:** We think both metrics are applicable following our observation in the paper, even for nonscalar functions (eg predicting trajectories of the dynamical system). However, the first measure introduced in Equation 12 theoritaclly could have more stability due to the average over the function $f$.
>
> **Motion Capture task:** For our algortihm, we sample from the SE(3) symmetry group in all the exeriments and not the subgroup for a fair comparison against the equivariant baselines.  Furthermore, we have added new results on the Motion Capture task with a comparison to approximate equivariance baselines. We compare our method against Residual Pathway Priors (RPP) [2], Projection-Based Equivariance Regularizer (PER) [3], and equivariant MLP (EMLP) [4]. As these architectures are designed based on linear layers and MLP, we apply the augmented loss to standard MLP with a similar number of layers and parameters. Our new results confirm the applicability of our method to different architectures, including MLPs, CNNs, GNNs, and Transformers, across a wide range of benchmarks.
>
> **Minor errors:** We thank the reviewer for these points, we have corrected the points in the updated version.
>
> 1. Automatic Symmetry Discovery with Lie Algebra Convolutional Network Dehmamy et al., NeurIPS 2021
>
> 2. Residual Pathway Priors for Soft Equivariance Constraints. Finzi et al., NeurIPS 2021.
>
> 3. Regularizing Towards Soft Equivariance Under Mixed Symmetries.  Kim et al., ICML 2023.
>
> 4. A Practical Method for Constructing Equivariant Multilayer Perceptrons for Arbitrary Matrix Groups. Finzi et al., 2021.

---

> > ### Author Response · Authors · 2024-11-22
> >
> > **Questions**
> >
> > **Data Efficiency:** As we pointed out before, we apply a hard setting on N-body dynamical systems, where we train on $100$ samples and test on $5000$ samples. Our algorithm competes with equivariant architectures in both in-distribution and out-of-distribution settings (Figure 2 in the paper).
> >
> > **Loss Landscape:** We agreed with the reviewer on the limitations of using two directions to plot the shape of the loss around the minima. This still needs further study, and we plan to explore this direction in future work.
> >
> >
> > We sincerely hope that we have addressed the concerns of the reviewer satisfactorily in the revised version and would kindly ask the reviewer to update their score accordingly.

---

> > ### Comment · Reviewer_EQn6 · 2024-11-23
> > **Thank you for your response**
> >
> > Thank you for your modifications and answers to the questions. I appreciate your efforts to give the readers a better picture and clarify most of my doubts.
> >
> > 1. I do not understand what the authors meant by "Nonetheless, these directions cannot seamlessly leverage unconstrained architectures that do not bake symmetries into their design by simply altering the training protocol." since most of the papers attempt to leverage the unconstrained architectures (even pretrained ones) and converting them to appropriate symmetries.
> > 2. I apologize for asking the authors to compare with the incorrect paper (i.e., [10]). I wanted to refer to Sec 3.2 of [13]. Please answer this (i.e., the difference between equivariance promoting regularizer of [13] and your $\mathcal{L}_{equi}$), and thank you for understanding.
> > 3. As per the previous version, I don't seem to find an answer to "How does Equation 9 work (or make sense) when $f(x)$ is non-scalar?"
> > 4. There is an incorrect citation in line 302, where [13] is cited as using "pretrained models."
> >
> > I am happy with the authors' answers and their new manuscript version. I await the answers to the remaining points.

---

> > > ### Author Response · Authors · 2024-11-23
> > >
> > > We thank the reviewer for his time and feedback and for engaging with us in the discussion.
> > >
> > > - **''I do not understand what the authors meant by "Nonetheless, these directions cannot seamlessly leverage unconstrained architectures that do not bake symmetries into their design by simply altering the training protocol." since most of the papers attempt to leverage the unconstrained architectures (even pretrained ones) and converting them to appropriate symmetries.''** We apologize for our ambiguous statement in the previous version of the paper. Our intention was to highlight that, despite methods like canonicalization and frame-averaging utilizing unconstrained architectures, these methods often require additional network or averaging techniques to achieve equivariance and may not rely solely on adjustments to the training protocol (e.g.,  loss function or optimization strategy) of unconstrained models. We have clarified this in an updated version of our manuscript.
> > >
> > > - **Comparison with [1]:** The equivariance-promoting regularizer defined in [1] injects equivariance in the latent space of the network by explicitly modeling transformations with an additional map, as it is not obvious how to apply the transformation group to latent vectors. The authors introduced a learnable map $M_{a}$ that is applied to the output of the encoder, which is a latent vector. The learnable map depends on the augmentation parameters, and the authors mentioned that for each augmentation (e.g. single rotation) they need a new map, which might be computationally problematic when doing a large number of augmentations. In our method, we learn the equivariance loss directly using the true label $y$ and apply the group action to it (e.g. for predicting the 3D positions of a system given the initial state,  we rotate $y$ by the same rotation matrix applied to the input $x$). Furthermore, we didn't have any additional parameters for the equivariance loss, and it is directly utilized in the optimization of the unconstrained model, making it computationally efficient. We have indicated the difference in the updated version.
> > >
> > > - **Non-scalar function:**  We apologize for missing this point. We have updated the paper and indicated that $\| \| \cdot \| \|_2$ is an $L_2$ norm, making it valid for non-scalar functions. The operator $\rho(g_i)$ represents the action of the group element $g_i$ on the output space of $f$. For example in vector-valued outputs, $\rho(g_i)$ is a linear operator (e.g., a rotation matrix) acting on the vector $f(x)$. Please note that equation 9 in the original submission is now equation 12 in the revised version of the manuscript.
> > >
> > > - **Incorrect citation:**  We thank the reviewer for this point and apologize for that. We have updated the paper with the corrected explanation.
> > >
> > >
> > > 1. Steerable Equivariant Representation Learning.  Bhardwaj et al., 2023

---

> > > > ### Comment · Reviewer_EQn6 · 2024-11-26
> > > > **Thank you for your response**
> > > >
> > > > Thank you for your answers and patience in clarifying my doubts. I believe the paper has improved and connects better with the existing literature. I have increased my scores to recommend acceptance. However, I do share similar concerns with other reviewers, particularly the limited novelty and applicability. I am a bit concerned that we would have to train different models with multiple combinations of $\alpha$ and $\beta$ for a particular task. It would be interesting to see more insights from the loss landscape and understand if this method can be applied during fine-tuning. For instance, you have a pretrained model, and you want to use it for a task which needs some extent of equivariance - can this method leverage the zero-shot loss landscape of the pretrained model?

---

> > > > > ### Author Response · Authors · 2024-11-26
> > > > >
> > > > > We thank the reviewer for their time and feedback, and for engaging with us in the discussion.
> > > > > We appreciate the reviewer’s acknowledgment of the revisions made and the improvement of the paper.
> > > > >
> > > > > In this work, we undertake a comprehensive comparison between existing equivariant models and their unconstrained counterparts across a diverse set of benchmarks. Specifically, we evaluate Transformers, Graph Neural Networks (GNNs), Convolutional Neural Networks (CNNs), and Multi-Layer Perceptrons (MLPs) on four distinct tasks: Dynamical Systems, Motion Capture, Molecular Dynamics, and Jet Flow benchmarks. We aim to provide valuable insights into the performance, scalability, and applicability of unconstrained vs equivariant models across various domains.
> > > > >
> > > > > However, we acknowledge that numerous additional ideas for extending our study offer exciting opportunities for future research.
> > > > > For example, as we indicated before, $\alpha$ and $\beta$ serve as additional hyperparameters in the training procedure.
> > > > > Future directions could explore the utilization of efficient learnable weights, such as [1], or recent approaches that use gradient projection as suggested in [2].
> > > > > Additionally, investigating the application of our method during the fine-tuning phase, such as leveraging pretrained models for tasks requiring equivariance, is an exciting prospect. For example, integrating our framework with denoising objectives [3, 4] could enhance its applicability and performance in scenarios where pretrained models are adapted to new tasks.
> > > > > We believe that this direction could significantly impact the field by enabling broader applicability and easier integration into existing frameworks.
> > > > >
> > > > >
> > > > > 1. SLAW: Scaled Loss Approximate Weighting for Efficient Multi-Task Learning. Crawshaw et al, 2021.
> > > > >
> > > > > 2. Task Weighting through Gradient Projection for Multitask Learning. Bohn et al., 2024.
> > > > >
> > > > > 3. Pre-training via Denoising for Molecular Property Prediction. Zaidi et al., ICLR 2023.
> > > > >
> > > > >
> > > > > 4. Pre-training with fractional denoising to enhance molecular property prediction. Ni et al., Nat Mach Intell 2024.

---

### Official Review · Reviewer_JGuC · 2024-11-03

**Soundness:** 2
**Presentation:** 2
**Contribution:** 2
**Rating:** 5
**Confidence:** 4

**Summary:**

The paper introduces an augmented loss function that incorporates a measure of average equivariance, aiming to enhance the prediction of approximately equivariant information. The authors validate this approach on both equivariant and non-equivariant tasks, utilizing transformer and graph neural network architectures. Furthermore, they conduct a visual analysis of the loss landscape, comparing two different architectures: Transformer with the augmented loss and GATr without.

**Strengths:**

1. The augmented loss function is generalizable across various architectures.

2. The augmented loss function requires relatively few samples to work effectively making it computationally efficient.

**Weaknesses:**

The proposed methodology can be described as approximate equivariance, but the paper lacks an adequate background and comparative analysis against existing works on approximate equivariance. This raises concerns about both the novelty and the empirical validation of the approach.

1.  Novelty: Augmented loss functions enforcing approximate equivariance have been studied (e.g. [1]) including an average measure (e.g. [2]).

2. Empirical Support:  The paper does not benchmark against other methods that address approximate equivariance (e.g., [1]), nor does it consider theoretically grounded approaches to symmetry breaking (e.g., [3], [4]) or simpler strategies like combining SE3Transformer with MLPs.

[1] Kim, Hyunsu, Hyungi Lee, Hongseok Yang, and Juho Lee. "Regularizing towards Soft Equivariance under Mixed Symmetries." Proceedings of the 40th International Conference on Machine Learning, ICML'23, 2023, pp. 686, JMLR.org.

[2]  K. Lin, B. Huang, L. M. Collins, K. Bradbury and J. M. Malof, "A simple rotational equivariance loss for generic convolutional segmentation networks: preliminary results," IGARSS 2019 - 2019 IEEE International Geoscience and Remote Sensing Symposium, Yokohama, Japan, 2019.

[3] Wang, Rui, Robin Walters, and Tess Smidt. "Relaxed Octahedral Group Convolution for Learning Symmetry Breaking in 3D Physical Systems." NeurIPS 2023 AI for Science Workshop, 2023, https://openreview.net/forum?id=B8EpSHEp9j.

[4] Lawrence, Hannah, Vasco Portilheiro, Yan Zhang, and Sékou-Oumar Kaba. "Improving Equivariant Networks with Probabilistic Symmetry Breaking." ICML 2024 Workshop on Geometry-grounded Representation Learning and Generative Modeling, 2024, https://openreview.net/forum?id=1VlRaXNMWO.

**Questions:**

1. To address fundamental concerns, I recommend a more comprehensive background and analysis, alongside broader empirical comparisons. Specifically, the study should include a wider range of architectures that go beyond strictly equivariant and non-equivariant models, particularly for the motion capture task, which is inherently non-equivariant.

2. In practice, if a single randomly sampled group element is used per sample in each training step (as mentioned at the end of Section 3.1), this should be explicitly stated in Section 3.2 where the sampling procedure is discussed and the number of samples M is introduced.

3.  The paper lacks heuristic, theoretical and/or empirical justification for the choice of one group element per sample per training epoch sufficient. As a result the particular choice of measure is understudied.  Moreover, it remains unclear how the equivariance error is measured in Section 6. How many samples are used for this computation?

4.  It's entirely unclear if the difference in the loss landscape is a result of the augmented loss function or a result of the architectural difference. I would strongly recommend performing comparisons with a fixed architecture.

5. The MD17 dataset includes two regression targets: energy and forces. Please clarify in the text which target is being used (likely force regression) and how the results are generated.

6. Near the end of Section 6.2, the statement "Best performance is observed at an intermediate level of equivariance..." is confusing. Since the paper modifies the loss function, not the architecture, this needs further explanation for supporting the proposed methodology. Otherwise, the conclusion is simply to not utilize strict equivariant architectures for non-equivariant tasks.

---

> ### Author Response · Authors · 2024-11-22
>
> We thank the reviewer for their time and all the comments, which have helped us improve our paper and indicate our contribution. We are also open to answer any further questions.
>
> **Novelty, and comparison to approximate equivariance:**
>
> - Our method is a general training procedure that could be applied to any unconstrained models, without specific assumption of the design (eg GNN/ CNN/ MLP or Transformers) while most of the work on approximate equivariance focus on relaxing equivariant architectures or regularizing specific layers within such architectures. We have added a discussion on this in the Related Work section with red colors in the updated manuscript.
>
> - The method defined in [1] is a regularization for the linear layers of Equivariant MLP [2], and [3]  regularized the CNN architecture. However, in our work, we didn't assume a specific class of models, which increases the applicability of our method to various domains. We have clarified this in the updated version.
>
> **Empirical evaluation:**
>
> - We have added a new comparison on Motion Capture task: We compare our method against Projection-Based Equivariance Regularizer (PER) [1], equivariant MLP (EMLP) [2], and Residual Pathway Priors (RPP) [4]. As these architectures are designed based on linear layers and MLP, we apply the augmented loss to standard MLP with a similar number of layers and parameters.
>
> - We have included a new task on the Jet Flow dataset used by [5] (a two-dimensional benchmark that captures turbulent velocity fields): We apply our method to a Convolutional neural network (CNN) and compare it with Relaxed Steerable Convolution (RSteer) [5] and E2CNN [6] (more details in Appendix A in the paper).
>
> - Our new results confirm the applicability of our method to different architectures, including MLPs, CNNs, in addition to GNNs, and Transformers, across a wide range of benchmarks.
>
> 1. Regularizing Towards Soft Equivariance Under Mixed Symmetries.  Kim et al., ICML 2023.
>
> 2. A Practical Method for Constructing Equivariant Multilayer Perceptrons for Arbitrary Matrix Groups. Finzi et al., 2021.
>
> 3. A simple rotational equivariance loss for generic convolutional segmentation networks: preliminary results.
>
> 4. Residual Pathway Priors for Soft Equivariance Constraints. Finzi et al., NeurIPS 2021.
>
> 5. Approximately Equivariant Networks for Imperfectly Symmetric Dynamics. Wang et al., ICML 2022.
>
> 6. General E(2)-Equivariant Steerable CNNs. Weiler et al., NeurIPS 2019.

---

> > ### Author Response · Authors · 2024-11-22
> >
> > **Questions:**
> >
> > - **Approximate equivariance:** We have included in the updated version new results comparing to the approximate equivariance baselines.
> >
> > - **Equivariance measure and augmented loss:**  There appears to be some misunderstanding between the equivariance measure and the augmented loss. For the augmented loss, we use a single random rotation during training, this is indicated in Section 3.1 where we defined the equivariance loss and the penalty parameters. Empirically, this is enough to achieve comparable performance with the equivariance baselines. Where the equivariance measure introduced in Section 4 quantify the degree of equivariance exhibited by a function $f$. We use this measure to compare different models and baselines in the experiments and how increasing the weights on the equivariance loss could reduce the equivariance error. For the equivariance measure, we use $M=100$ samples from the group and noticed this was sufficient to obtain stable results. We clarified all the points in the updated version.
> >
> > - **Justification for the choice of one group element per sample:** We conduct additional experiments with different number of samples from the symmetry group during training comparing our method and data augmentation. Our new results confirm that we can acheive reasanble performance using fewer samples from the symmetry group. We included the results in Appendix C.2 in the updated version.
> >
> >
> > - **The equivariance measure** used in Section 6 is defined in Equation 13. We explained this at the beginning of Section 6 in the updated version.
> >
> > - **Loss landscape:** We would like to emphasize that unconstrained models may exhibit a more convex or smoother structure around their local minima compared to equivariant models. This observation could serve as additional evidence of the optimization challenges faced by equivariant networks. However, we acknowledge certain limitations in this analysis, such as not accounting for the trajectories each model follows to reach their respective minima. We have clarified this point in the Limitations section and plan to explore it further in future work.
> >
> > - **MD17** The task in MD17 is predicting the 3D trajectories of molecules, similar to [1, 2]. We have provided a detailed explanation of this in Section 6.3.
> >
> > - **The statement**  "Best performance is observed at an intermediate level of equivariance...": This statement refers to the observation that adjusting the weighting parameters of the augmented loss allows us to control the level of approximate equivariance learned by the model (using the quantified measure as in Figures 2, 3, and 4 in the paper). We noticed there is a trade-off between equivariance and performance, where the best performance could be achieved with a lower weight in the equivariance loss. We have clarified this in the updated version.
> >
> > 1. Equivariant graph mechanics networks with constraints. Huang et al., 2022.
> >
> > 2. Equivariant Graph Neural Operator for Modeling 3D Dynamics
> > Xu et al., ICML 2024.
> >
> > We sincerely hope that we have addressed the concerns of the reviewer satisfactorily in the revised version and would kindly ask the reviewer to update their score accordingly.

---

> > > ### Comment · Reviewer_JGuC · 2024-11-25
> > > **Response to Authors**
> > >
> > > I thank the authors for their response and the revisions which have improved the paper. I have updated my score accordingly.
> > >
> > > I find that the paper still has significant weaknesses. The MD17 task presented is still unclear, and the results are from a procedure that limits the range of baselines. There are MD17 baselines that consider significantly broader range of baselines with better results than those reported. Furthermore, I find the significance of the contribution to be low due to previous studies of the loss function.

---

> > > > ### Author Response · Authors · 2024-11-26
> > > >
> > > > We thank the reviewer for their time and for engaging with us in the discussion. We appreciate the reviewer’s acknowledgment of the revisions made and the improvement of the paper.
> > > >
> > > > Regarding the MD17 benchmark, we would like to clarify that it has two common tasks in the literature:
> > > >
> > > > - Invariant Task: Predicting energies given molecular states/ positions (e.g. [1, 2, 3, 4]).
> > > > - Equivariant Task: Predicting molecular states/ positions after a specific number of time steps given initial states/ positions (e.g. [5, 6, 7, 8]).
> > > >
> > > >
> > > > In this work, we focus on the equivariant task, following previous work on this task.  Our primary objective is to compare unconstrained models with their corresponding equivariant versions (GNN vs EGNN).
> > > >
> > > > Furthermore, we have expanded the related work section to include a detailed discussion of the existing literature on approximate equivariance, with all the differences between these approaches and our proposed method.
> > > > If there are any further questions or specific aspects that we have not addressed, we are happy to provide additional clarifications.
> > > >
> > > >
> > > > 1. SchNet: A continuous-filter convolutional neural network for modeling quantum interactions. Schütt et al., NeurIPS 2017.
> > > >
> > > > 2. Rotation Invariant Graph Neural Networks using Spin Convolutions. Shuaibi1 et al.,  2021.
> > > >
> > > > 3. Spherical Message Passing for 3D Graph Networks. Liu et al., ICLR 2022.
> > > >
> > > > 4. Symmetry-Informed Geometric Representation for Molecules, Proteins, and Crystalline Materials. Liu et al., NeurIPS 2023.
> > > >
> > > > 5. Equivariant graph mechanics networks with constraints. Huang et al., ICLR 2022.
> > > >
> > > > 6. EqMotion: Equivariant Multi-Agent Motion Prediction With Invariant Interaction Reasoning. Xu et al., CVPR 2023.
> > > >
> > > > 7. Equivariant Spatio-Temporal Attentive Graph Networks to Simulate Physical Dynamics. Wu et al., NeurIPS 2023.
> > > >
> > > > 8. Equivariant Graph Neural Operator for Modeling 3D Dynamics. Xu et al., ICML 2024.

---

> > > > > ### Author Response · Authors · 2024-12-02
> > > > > **Follow Up**
> > > > >
> > > > > Dear reviewer,
> > > > >
> > > > > Thank you once again for your time and the valuable feedback you have provided. We did our best to answer your questions and follow your comments in the revised version.
> > > > > As the discussion period is ending soon, we kindly ask if the reviewer can consider updating their score, or if they have any further questions.

---

### Official Review · Reviewer_eYaK · 2024-11-04

**Soundness:** 3
**Presentation:** 3
**Contribution:** 2
**Rating:** 5
**Confidence:** 3

**Summary:**

The paper investigates unconstrained models for handling data symmetry. The authors demonstrate that by designing a loss function specifically tailored for learning equivariance, unconstrained models can approximate data symmetry by minimizing this equivariant loss. This approach allows the models to efficiently control the level of equivariance while maintaining flexibility.

**Strengths:**

- The paper is well-written and presents the core ideas in a clear and accessible manner.
- Using a "landscape" to describe the benefits of unconstrained models is particularly novel and insightful.

**Weaknesses:**

- The study introduces this equivariant loss without providing a strong theoretical foundation for the proposed approach. Also, the proposed method is quite straightforward, and its distinction from data augmentation is unclear. It essentially computes the loss on a larger augmented dataset by sampling transformed data.   I suspect this method is already well-known within the community, which limits the novelty of the contribution.

- The experimental comparisons are performed on a limited set of classic models rather than state-of-the-art models, raising concerns about the practical applicability of the method to more advanced techniques.

**Questions:**

Please address my concerns in the weakness part, especially the novelty of the proposed method and the theoretical foundations.

---

> ### Author Response · Authors · 2024-11-22
>
> We thank the reviewer for their time and comments, which have helped us improve our paper. We are also open to answer any further questions.
>
>
> **Theoretical motivations and differences from data augmentation:**
>
> - Our theoretical motivation is that by having an adaptive parameter $\beta$ on the equivariance loss, we can modulate the extent to which a model exhibits equivariance, depending on the requirements of the task. In Section 4, we introduce a measure that quantifies the level of learned equivariance in the model which we use to analyze our results.
>
> - The original difference from data augmentation is that we utilize an additional controlled equivariance loss together with the objective loss that both minimized during training. We consider two distinct approaches to regulate the penalty parameters $\alpha$ and $\beta$: constant penalty and gradual penalty. For
>     constant penalty, we assign a fixed weight to each task’s loss throughout the training process. In contrast,
>     the gradual penalty dynamically adjusts the weights of each task’s loss during training. For gradual penalty, we use GradNorm algorithm [1], which is particularly
>     suited for tasks that involve simultaneous optimization of multiple loss components, as it dynamically adjusts the weight of each loss during training. We clarified this in Section 3.2 of the paper.
>
> - By minimizing the equivariance loss term simultaneously with the objective function, we can control the equivariance objective depending on the parameter $\beta$. This allows us to systematically adjust the degree of equivariance the model learns (Section 6.1 in the paper).
>
> - We noticed this decomposition is important to control the trade-off between equivariance and performance for multiple tasks (Motion Capture in Sections 6.2 and Molecular Dynamics in Section 6.3 of the paper)
>
> - Data augmentation can also be viewed as a special case of our method with $\alpha = 0$ and $\beta = 1$. We clarified this in Section 3.3 of the paper.
>
> **Comparing to state of the art:**
>
> In this work, we consider Transformers and Graph Neural Networks (GNNs), with their equivariant versions, as our main baselines.
>
> - We compared Transformers against SE(3)-Transformer [2], and Geometric Algebra Transformer [3] which is a recent equivariant architecture for geometric data (Sections 6.2 and 6.3 in the paper).
>
> - We consider another comparison between GNN and EGNN for molecular dynamics tasks (Section 6.3).
>
> - While we have added new results on MLPs and CNNs architectures, we think this will be a future direction to apply our augmented loss to a broader range of unconstrained models.
>
> We sincerely hope that we have addressed the concerns of the reviewer satisfactorily in the revised version and would kindly ask the reviewer to update their score accordingly.
>
> 1. GradNorm: Gradient Normalization for Adaptive Loss Balancing in Deep Multitask Networks. Chen et al., ICML 2018.
>
>
> 2. SE(3)-Transformers: 3D Roto-Translation Equivariant Attention Networks. Fuchs et al., NeurIPS 2020.
>
>
> 3. Geometric Algebra Transformer. Brehmer et al., NeurIPS 2023.

---

> > ### Comment · Reviewer_eYaK · 2024-11-24
> > **Response to Authors**
> >
> > Thank you for addressing my concerns regarding the novelty of the proposed methods and for incorporating new content based on all reviewers' feedback. The additions in the revision are helpful and clarify the paper's contributions. I have raised my scores accordingly.
> >
> > However, the overall idea and methods, while interesting, are relatively straightforward, which limits the paper's contribution. The concept of the landscape is quite interesting, and I believe that further theoretical investigation and empirical studies in this direction could significantly strengthen the paper and elevate its impact.

---

> > > ### Author Response · Authors · 2024-11-26
> > >
> > > We thank the reviewer for their time and for engaging with us in the discussion. We appreciate the reviewer's acknowledgment of the revisions made in response to all reviewers' feedback, and that the reviewers found our work and the use of the landscape concept interesting.
> > >
> > > In this study, we undertake a comprehensive comparison between existing equivariant models and their unconstrained counterparts across a diverse set of benchmarks. Specifically, we evaluate Transformers, Graph Neural Networks (GNNs), Convolutional Neural Networks (CNNs), and Multi-Layer Perceptrons (MLPs) on four distinct tasks: Dynamical Systems, Motion Capture, Molecular Dynamics, and Jet Flow benchmarks. We aim to provide valuable insights into the performance, scalability, and applicability of unconstrained vs equivariant models across various domains.
> > >
> > > We believe that simple approaches capable of understanding and analyzing unconstrained versus equivariant models significantly impact the field by enabling broader applicability and easier integration into existing frameworks. Finally, we acknowledge that numerous additional ideas for extending our study offer exciting opportunities for future research.

---

> > > > ### Author Response · Authors · 2024-12-02
> > > > **Follow Up**
> > > >
> > > > Dear reviewer,
> > > >
> > > > Thank you once again for your time and the valuable feedback you have provided. We did our best to answer your questions and follow your comments in the revised version.
> > > > As the discussion period is ending soon, we kindly ask if the reviewer can consider updating their score, or if they have any further questions.

---

### Author Response · Authors · 2024-11-22
**Response to all reviewers and ACs**

We thank all the reviewers for their time and valuable feedback, which have helped improve our paper and confirm our contributions.

The reviewers have highlighted several strengths of our work:

- eYaK: "The paper is well-written and presents the core ideas in a clear and
    accessible manner. Using a "landscape" to describe the benefits of unconstrained models is particularly novel and insightful."
- JGuC: "The augmented loss function is generalizable across various architectures.
    The augmented loss function requires relatively few samples to work effectively making it computationally efficient."
- EQn6: "The experiments are conducted in different domains and examine several
    essential aspects of the algorithm, giving more insights into the method
    and how levels of equivariance can affect downstream task performance."
- JQbk: "Relaxing equivariance is a valuable research direction that can break
    through the constraints on generalization or expressive power caused by
    strictly equivariant operations."

Our work considers the current active topic on learning symmetries in unconstrained models versus constrained equivariant models. We consider two general architectures, Graph Neural Networks (GNNs) and Transformers, along with their equivariant versions. We test our approach on three different tasks: N-body dynamical system, Motion Capture, and Molecular Dynamics.

However, we found that most of the concerns mentioned by reviewers focus on additional evaluation and analysis compared to the prior work on approximate equivariance. Specifically, there are concerns about the Motion Capture task not being fully E(3) equivariance. We have updated the manuscript (changes highlighted in red color) following their feedback. The updated version includes:

- A new section discussing approximate equivariance and included comparisons to relevant prior work.

- A new comparison on Motion Capture task: We compare our method against Residual Pathway Priors (RPP) [1], Projection-Based Equivariance Regularizer (PER) [2], and equivariant MLP (EMLP) [3]. As these architectures are designed based on linear layers and MLP, we apply the augmented loss to standard MLP with a similar number of layers and parameters.

**Table: Performance on Motion Capture dataset (MSE × 10⁻²)**

|            | EMLP | RPP | PER | MLP  | Data Augment. | Ours
|----------------------|------------|--------------|------------|--------------|------------|--------------|
| **Walking (Subject #35)**          | 7.01 ± 0.46      | 6.99 ± 0.21      | 7.48 ± 0.39        | 6.80 ± 0.18    | 6.37 ± 0.04         | **6.04 ± 0.09**     |
| **Running (Subject #9)**           | 57.38 ± 8.39      | 34.18 ± 2.00      | 33.03 ± 0.37       | 39.56 ± 2.25    | 40.23 ± 0.94       | **32.57 ± 1.47**   |

- A new task on the Jet Flow dataset used by [4] (a two-dimensional benchmark that captures turbulent velocity fields): We apply our method to Convolutional neural network (CNN) and compare it with Relaxed Steerable Convolution (RSteer) [4] and E2CNN [5] (more details in Appendix A in the updated version).


**Table: Performance on Jet Flow dataset (RMSE)**

| Model   | Future               | Domain               |
|---------|----------------------|----------------------|
| E2CNN   | 0.21 ± 0.02          | 0.27 ± 0.03          |
| RSteer  | *0.17 ± 0.01*        | **0.16 ± 0.01**      |
| Ours    | **0.16 ± 0.003**     | *0.18 ± 0.003*       |


Our new results confirm the applicability of our method to different architectures, including MLPs, CNNs, GNNs, and Transformers, across a wide range of benchmarks. We thank the reviewers again for their valuable feedback and kindly ask them to consider increasing their scores if that addresses their concerns.



1. Residual Pathway Priors for Soft Equivariance Constraints. Finzi et al., NeurIPS 2021.

2. Regularizing Towards Soft Equivariance Under Mixed Symmetries.  Kim et al., ICML 2023.

3. A Practical Method for Constructing Equivariant Multilayer Perceptrons for Arbitrary Matrix Groups. Finzi et al., 2021.

4. Approximately Equivariant Networks for Imperfectly Symmetric Dynamics. Wang et al., ICML 2022.

5. General E(2)-Equivariant Steerable CNNs. Weiler et al., NeurIPS 2019.

---

### Comment · Area_Chair_3gmH · 2024-11-25

Dear reviewers,

If you haven’t done so already, please engage in the discussion as soon as possible. Specifically, please acknowledge that you have thoroughly reviewed the authors' rebuttal and indicate whether your concerns have been adequately addressed. Your input during this critical phase is essential—not only for the authors but also for your fellow reviewers and the Area Chair—to ensure a fair evaluation.
Best wishes,
AC

---

### Meta-Review · Area_Chair_3gmH · 2024-12-21

**Metareview:**

This paper introduces a framework for learning approximate symmetries from data by starting with a neural network that is not inherently equivariant and training it with an equivariance regularizer. The authors made commendable efforts to address several concerns raised by the reviewers, such as the lack of comparisons with key existing works and the need to demonstrate the framework's effectiveness on large-scale, complex systems. However, despite these improvements, the reviewers remain unenthusiastic about the paper. Their main reservations lie in its limited novelty (similar approaches, such as Kim et al., 2023, have already been explored, albeit with a focus on linear layers) and its restricted applicability (e.g., requiring multiple training runs with varying $\alpha$ and $\beta$ values). The AC encourages the authors to further refine this research direction and address the outstanding issues regarding the framework's applicability and practicality.

**Additional Comments On Reviewer Discussion:**

The initial reviews were quite bad, and the authors resolved some of the concerns by clarifying misunderstandings and adding more empirical results. Still, only one reviewer raised the score above the acceptance bar (to weak accept), since the fundamental weakness (limited novelty and applicability) has not been fully addressed. None of the reviewers were willing to champion the paper during the final discussion phase.

---

### Decision · Program_Chairs · 2025-01-22

Reject